# Anabolic Resistance in the Pathogenesis of Sarcopenia in the Elderly: Role of Nutrition and Exercise in Young and Old People

**DOI:** 10.3390/nu15184073

**Published:** 2023-09-20

**Authors:** Caterina Tezze, Marco Sandri, Paolo Tessari

**Affiliations:** 1Department of Biomedical Sciences, University of Padova, via Ugo Bassi 58/b, 35121 Padova, Italy; marco.sandri@unipd.it; 2Veneto Institute of Molecular Medicine, via Orus 2, 35129 Padova, Italy; 3Department of Medicine, McGill University, Montreal, QC H4A 3J1, Canada; 4Department of Medicine, University of Padova, via Giustiniani 2, 35128 Padova, Italy

**Keywords:** exercise, intracellular signals, nutrition, protein foods, protein synthesis, sarcopenia, skeletal muscle

## Abstract

The development of sarcopenia in the elderly is associated with many potential factors and/or processes that impair the renovation and maintenance of skeletal muscle mass and strength as ageing progresses. Among them, a defect by skeletal muscle to respond to anabolic stimuli is to be considered. Common anabolic stimuli/signals in skeletal muscle are hormones (insulin, growth hormones, IGF-1, androgens, and β-agonists such epinephrine), substrates (amino acids such as protein precursors on top, but also glucose and fat, as source of energy), metabolites (such as β-agonists and HMB), various biochemical/intracellular mediators), physical exercise, neurogenic and immune-modulating factors, etc. Each of them may exhibit a reduced effect upon skeletal muscle in ageing. In this article, we overview the role of anabolic signals on muscle metabolism, as well as currently available evidence of resistance, at the skeletal muscle level, to anabolic factors, from both in vitro and in vivo studies. Some indications on how to augment the effects of anabolic signals on skeletal muscle are provided.

## 1. Introduction

The term “sarcopenia” (from the Greek words “sarx”, i.e., flesh, and “penia”, i.e., loss) was first proposed by Rosenberg in 1989 to describe the age-related loss of muscle mass [1]. Later, this term referred to the decrease in muscle mass and/or strength with ageing. Since 2016, the World Health Organization (WHO)’s International Statistical Classification of Diseases and Related Health Problems (ICD) has recognized sarcopenia as a disease (code ICD-10-CM) (M62.84) [2]. Sarcopenia, defined as an age-related loss in skeletal muscle mass and function, is one of the most pressing health problems in the elderly. Depending on the definition used for sarcopenia, the prevalence in 60- to 70-year olds is reported to be between 6% and 22%, while the prevalence is as high as 50% in people > 80 years old [2]. 

There are several definitions of sarcopenia. The three widely recognized ones are from the following International Working Groups or Societies [2,3,4]:

The European Working Group on Sarcopenia in Older People (EWGSOP): They defined sarcopenia in 2010 as “a syndrome characterized by progressive and generalized loss of skeletal muscle mass and strength, with a risk of adverse outcomes such as physical disability, poor quality of life, and death”.

The International Working Group on Sarcopenia (IWGS): The IWGS proposed a definition in 2011: “Sarcopenia is defined as a syndrome characterized by progressive and generalized loss of skeletal muscle mass and strength, and it is strictly correlated with physical disability, poor quality of life, and death”.The Foundation for the National Institutes of Health (FNIH): In collaboration with various experts, they established a working definition in 2014: “Low muscle mass as measured by any method, and low muscle strength and/or low physical performance”.Definition and Outcomes Consortium’s (SDOC) efforts, which involved a literature review, data analysis from multiple studies, and expert panel review to establish an evidence-based understanding of sarcopenia. They define sarcopenia as characterized by both weakness (defined by low grip strength) and slowness (defined by low usual gait speed) in older adults. These two components are considered important discriminators and predictors of various adverse health-related outcomes, including falls, self-reported mobility limitation, hip fractures, and mortality in community-dwelling older adults.

As the world’s older adult population increases, so does the number of individuals with sarcopenia. To ensure effective treatment, clinicians and operators must accurately measure sarcopenia before initiating treatment or interventions. It is essential to use an objective measurement tool for diagnosis and recommend utilizing any of the validated options we reported upstream.

Sarcopenia is an almost unavoidable process associated with ageing, albeit occurring at variable rates and different magnitudes. A variety of factors condition both its progression rate and the ultimate extent of the functional impairment [3,4]. Muscular loss has been estimated to occur at a rate of 1–3% yearly [5]. Besides sarcopenia, the overall impairment of neuromuscular function, defined as “dynapenia”, i.e., muscle weakness, is recognized as a practical, critical factor leading to the loss of physical independence in the elderly [4]. Loss of muscle fiber number is one of the main causes of sarcopenia, although fiber atrophy—particularly among type II fibers—is also involved [6,7]. Type II muscle fibers seem to play an important role in the ageing process of human skeletal muscles. According to this literature, loss of fibers, decrease in size, and fiber-type grouping represent major quantitative changes [8].

Risk factors commonly associated with the development of sarcopenia with ageing are a progressive decrease in physical activity; the coexistence of subtle or overt malnutrition and of age-related diseases; a higher body mass index with increased visceral fat area; smoking; sleep duration and poor quality; pharmacological therapies; and a chronic (sub)-inflammatory condition [9,10,11]. Each of these factors may impair muscle mass, function (i.e., strength, speed of contraction, and overall power), or both. These factors interact with each other, so that often it cannot be entirely clear which one comes first. A subjective reaction to perceived fatigue may also lead to an involuntary reduction in daily physical activity, thus amplifying muscle disuse in a negative loop. The mechanisms(s) for the increased fatigability occurring with ageing are incompletely understood [12].

Maintenance, wasting, and recovery of muscle mass depend on the relative rates of two opposite, ongoing processes, i.e., protein degradation (PD) and protein synthesis (PS), in other words, of protein turnover. These two processes occur simultaneously although at variable rates, and their regulation depends on different factors, operating in a complex network. Therefore, net protein anabolism (or “accretion”), i.e., the net increase in muscle mass, occurs anytime when synthesis exceeds degradation.

Nutrition and exercise are two major factors involved in the development of sarcopenia in the elderly. The aim of this narrative review is to provide an update of the current literature, addressing the role of (protein) nutrition and physical exercise, both as possible factors causing sarcopenia when defective, and as tools to prevent and/or to cure it when adequately provided. Due to the complex, sometimes contrasting body of existing data, we attempted to briefly and critically describe the experimental setting of each cited report, with the purpose of supporting each author’s conclusion. We provide also a review of the molecular mechanism(s) of muscle protein accretion, as well as the role of substrates and hormones on muscle protein turnover.

## 2. Part 1: Physiological Regulation of Muscle Metabolism and Growth

### 2.1. The Molecular Mechanisms behind Muscles Growth in Young Subjects: Exercise, Nutrients, and Hormones

Muscle growth, or muscle hypertrophy, is a complex process regulated by several molecular pathways. The primary pathways that promote muscle growth in response to exercise, nutrients, and growth signals include the IGF-1/PI3K (phosphoinositide 3-kinase)/Akt/mTOR pathway and hormones like testosterone and GH (Figure 1). 

The IGF-1/PI3K/Akt/mTOR pathway is a vital signaling cascade in muscle growth that involves various interconnected mechanisms. Its activation increases protein synthesis, reduces protein degradation, and improves cell growth. The molecular mechanisms and interactions within this pathway are still being studied in human muscle physiology. When activated, mTOR promotes muscle hypertrophy by stimulating protein synthesis and cell growth [13]. The mTOR kinase is the main regulator of muscle protein synthesis and responds to the availability of nutrients, particularly amino acids and growth factors. Indeed, resistance exercise and nutrient intake, especially leucine-rich amino acids, activate the mTOR pathway essential for exercise-induced muscle protein synthesis and are, thus, necessary for increasing muscle protein synthesis following resistance exercise in young men [14]. The key importance of mTOR is exemplified by an experiment, where young subjects were treated with a potent mTORC1 inhibitor (rapamycin) before performing a series of high-intensity muscle contractions to demonstrate this activation. Rapamycin treatment blocked the early, acute (after 1–2 h) contraction-induced increase in human muscle protein synthesis. Other studies have also shown that a high-protein diet enhances mTOR-mediated protein synthesis in young men following endurance exercise [15]. These studies indicate that mTOR activation is crucial in mediating the anabolic response to exercise and diet in humans (Figure 1).

Akt activation is crucial in promoting muscle protein synthesis in response to exercise and nutrient intake in young individuals [16]. In response to resistance training, significant hypertrophy (+10%) in the human quadriceps was demonstrated, with a parallel increase in phospho-Akt, phospho-GSK-3beta, and phospho-mTOR protein, and a decrease in the nuclear protein content of Foxo1, which is the master regulator of the atrophy program [16]. Akt is also influenced by nutrition, as excess leucine intake has been shown to enhance Akt signaling in young individuals, suggesting that Akt is sensitive to dietary amino acids and can modulate protein synthesis accordingly [17].

The timing of exercise and protein intake also affect Akt activation and subsequent muscle protein synthesis. While exercise alone did not increase Akt and mTOR phosphorylation, protein ingestion afterward did so in a dose-dependent manner. As a matter of fact, Akt activation is a complex process influenced by exercise, nutrition, and specific amino acids, and further research is being conducted to fully understand its role in muscle growth and adaptation [18].

The upstream controller in this axis is the insulin-like growth factor-1 (IGF-1), which is crucial in promoting growth and anabolic processes in skeletal muscles [13]. IGF-1 is a key growth factor that regulates both anabolic and catabolic pathways in skeletal muscle.

Studies indicate that IGF-1 induces hypertrophy of human myotubes in vitro, characterized by an increase in the mean number of nuclei per myotube, an increase in the fusion index, and an increase in myosin heavy chain (MyHC) content [19]. IGF-1 contributes to muscle protein turnover, protein synthesis, and the adaptation of skeletal muscles to resistance training [20]. Hormonal factors, such as ethinyl estradiol administration, can affect IGF-1 synthesis and degradation in skeletal muscles, potentially modulating muscle protein turnover and influencing responses to exercise and nutrient intake [21]. IGF1 is closely connected with growth hormone (GH), which stimulates muscle growth through several mechanisms and interactions with other signaling pathways. It stimulates protein synthesis by enhancing the uptake of amino acids into muscle tissue, providing the building blocks for muscle protein synthesis. This process is mediated, at least in part, by the GH/IGF-1 axis. GH stimulates the liver and other tissues to produce the insulin-like growth factor 1 (IGF-1), which promotes cell growth, protein synthesis, and the proliferation of satellite cells involved in muscle repair and growth [22].

GH promotes the uptake of essential nutrients, such as glucose and amino acids, into muscle cells for energy production and protein synthesis. It also affects metabolism by enhancing the breakdown of fats, providing additional energy sources for muscle growth [23]. However, pharmacological GH supplementation only increases muscle strength or size in individuals with clinical GH deficiency, and there is no evidence that exercise-induced changes in GH have the same effects in individuals with normal GH levels [24].

Testosterone is one of the most potent naturally secreted androgenic-anabolic hormones, and its biological effects include promoting muscle growth. In muscle, testosterone stimulates protein synthesis (anabolic effect) and inhibits protein degradation (anti-catabolic effect) [25,26,27,28,29]. Testosterone plays a crucial role in muscle growth in response to exercise and nutrition. Various studies have shed light on this topic. Vingren et al. (2010) explored the physiological aspects of testosterone in resistance exercise and training, highlighting its upstream regulatory elements [28]. They found that testosterone enhances muscle protein synthesis, stimulates satellite cell activation and proliferation, and modulates anabolic signaling pathways such as mTOR and IGF-1. These findings suggest that testosterone plays a key role in mediating the anabolic response to resistance exercise [28]. West and Phillips (2010) discussed the anabolic processes in human skeletal muscle, emphasizing the roles of growth hormone and testosterone (24). They concluded that testosterone acts directly on muscle tissue to promote muscle protein synthesis and hypertrophy.

The interactions between testosterone, growth hormone, and anabolic signaling pathways, such as mTOR and IGF-1, contribute to the overall anabolic response in muscle. West and Philips highlighted the importance of optimizing testosterone levels for maximizing muscle growth in response to exercise and nutrition [24]. In their work, the exercise paradigms are designed based on the assumption (not necessarily evidenced-based mechanisms) that GH and testosterone facilitate anabolic processes that lead to skeletal muscle protein accretion and hypertrophy. Furthermore, they pointed out that exercise-induced hormonal stimulation does not enhance intracellular markers of anabolic signaling or the acute post-exercise elevation of myofibrillar protein synthesis. Furthermore, they demonstrated that exercise-induced increases in GH and testosterone availability are unnecessary and do not enhance strength and hypertrophy adaptations. So, they concluded that local mechanisms, intrinsic to the skeletal muscle tissue performing the resistive contractions (i.e., weight lifting) are predominant in stimulating anabolism [24]. The regulation of satellite cells following myotrauma caused by resistance exercise is also related to testosterone, which plays a critical role in activating satellite cells, which are crucial for muscle repair and growth [29]. Finally, Bhasin et al. (2001) examined the dose–response relationships of testosterone in healthy young men, and they found that testosterone administration at varying doses increased dose-dependently muscle protein synthesis rates, resulting in greater muscle mass and strength gains [30]. These findings suggest that higher testosterone levels promote anabolic processes and contribute to muscle growth in response to exercise and nutrition

### 2.2. Effects of Substrates and Exercise on Skeletal Muscle Protein Synthesis in Young, Middle-Aged Subjects 

Accretion of muscle mass depends on physiological, metabolic, and hormonal factors, as well as on physical activity. Conversely, it is hindered by inactivity, malnutrition, overt diseases, and/or subtle, chronic pathological conditions [31,32,33,34,35,36]. The main protein-anabolic factors, at both the whole-body and skeletal muscle level, are the proteins and/or the amino acids themselves (i.e., the protein building blocks), as well as physical activity/exercise. In addition, energy availability, anabolic hormones (insulin, human Growth Hormone (hGH), IGF-1, β-agonists, anabolic steroids, and *see also above*), adequate tissue perfusion, and, in general, a “healthy status” (i.e., the absence of both overt diseases and subtle pathological conditions, such as a chronic sub-inflammatory status) also condition skeletal muscle accretion. Furthermore, the effects of any of these factors could be divided into either “acute” (i.e., detectable under acute experimental conditions) or “chronic” (i.e., following repeated stimuli, with end-point results tested sometime after).

Protein anabolism is thus achieved through the stimulation of protein synthesis and/or the inhibition of protein degradation. In this narrative review, we will focus predominantly on the regulation of muscle protein synthesis (MPS). MPS can be determined in vivo either by measuring the incorporation of infused amino acid stable isotopes into muscle by biopsy (Figure 2) and/or through measurements of the A-V difference in labeled as well as unlabeled amino acid(s) across a sampled district predominantly constituted by skeletal muscle, typically a limb (either the leg or the forearm) [37] (Figure 3).

#### 2.2.1. Effects of Proteins and Amino Acids

Definitely, protein ingestion stimulates skeletal muscle protein synthesis (MPS) [38]. The stimulation’s magnitude and duration depend on both the protein dose and its type/quality, which is closely associated with the concurrent post-ingestion rise in amino acid plasma concentrations [39,40]. Hyperaminoacidemia [41,42,43], specifically that of essential amino acids (EAAs) [44], among them, of the branched chain ones and of leucine in particular [31,45,46], largely condition tissue protein synthesis. 

Leucine is particularly important as a key metabolic regulator of MPS through its activation of the mTOR pathway, and it acutely enhances skeletal MPS both in vitro [45,47,48,49,50] (also above) and in vivo [51]. In addition to the protein or the amino acids, other variables or factors, such as the coexistence of exercise, the pattern and/or the timing of nutrient administration, the age of subjects, the presence of comorbidities, etc., are important. Most reports have investigated the combined effect of protein and exercise. As an example, typing in a PubMed search the string, “Stimulation of muscle protein synthesis by protein ingestion” identified ≈ 600 papers. Conversely, typing, “Stimulation of muscle protein synthesis by protein ingestion and exercise” identified ≈ 250 papers, whereas the string, “Stimulation of muscle protein synthesis by protein ingestion at rest” identified 60 papers. Therefore, although there is a large overlap among these selections, such an example simply underlines the predominant interest in combining protein nutrition with exercise. 

In non-exercising young subjects (i.e., “at rest”), the acute administration of either a protein-containing mixed meal or a pure protein load, stimulated skeletal muscle protein synthesis [52,53]. In dose–response studies, intake of 20 g of high-quality proteins, i.e., whey protein [54] or mixed egg protein [55], was sufficient to maximally stimulate postabsorptive rates of myofibrillar MPS in resting young men over 4 h [54]. 

In middle-aged men however, following the ingestion of graded amounts of beef (from 57 g, i.e., 12 g protein; 113 g, i.e., 24 g protein; or 170 g, i.e., 36 g protein), the stimulation of myofibrillar MPS was the greatest with 170 g of beef [56], thus apparently not achieving a plateau. Notably, in this study, exercise further and significantly enhanced the beef protein effect only at the highest administered dose (170 g beef). Similarly, following the administration to resting young volunteers, of a mixed meal containing both animal (beef) and vegetal proteins, at protein intakes of either 40 g or 70 g, the stimulation of skeletal muscle protein synthesis was similar at both doses, also independent of prior resistance exercise, thus leading to the conclusion that the ≈ 40 g mixed protein dose may attain the maximum effect on MPS [57]. The (marginal) inconsistency between the above-referenced studies could be due, besides possible experimental variations, either to the type of the administered protein (pure whey protein, i.e., a high quality, fast absorbable protein, vs. either beef, or mixed animal and vegetal proteins), or to the complex protein matrix of “natural” proteins that could retain specific, yet unappreciated effects.

The administration of free amino acids, either as bolus ingestion of 15 g EAA [58], or of a leucine-rich EAA and carbohydrate mixture [59], increased human muscle protein synthesis too. When dose–response curves between crystalline EAA and MPS were constructed, the literature data somehow contrasted. It was initially reported that 2.5 g crystalline EAA was sufficient to elicit an increase above basal of MPS in young subjects [60]. However, in subsequent studies using intact whey protein, the lower dose capable of eliciting a response in MPS in young muscle was set at >10 g (=5 g EAA), reaching saturation at 20–40 g EAA (Table 1).

Leucine, isoleucine, and valine, i.e., the branched-chain amino acids (BCAA), account for about one-third of all amino acid residues in muscle protein, and have been extensively studied regarding their direct regulatory role (due to leucine) on protein synthesis (see also above). Nevertheless, the demonstration of a clear-cut effect of BCAA alone, on skeletal muscle hypertrophy (i.e., a long-term effect) in humans is not sound. In a meta-analysis, leucine supplementation was reported to increase the muscle protein fractional synthesis rate, however, without changing either body lean mass or leg lean mass [61]. The effect of another amino acid, glutamine (non-essential), on skeletal muscle in humans, is yet uncertain and/or unwarranted [62,63,64]. Intravenous glutamine did not stimulate mixed muscle protein synthesis in healthy subjects [65].

The type of the administered protein is important too. Since the amount of the EAAs, and/or their relative proportions, are greater and/or more balanced in high- than in low-quality proteins, the amount and quality of the protein are relevant in the stimulation of protein synthesis. This issue is important in ageing, because the estimated recommended daily protein intake in aged people is ≈50% greater (1.2 g/kg BW) than that of young-adult subjects (0.8 g/kg BW) [66], and it could be better achieved by the intake of high-quality protein, such as whey protein, albumin, egg, or, to a lesser extent, of mixed milk protein, thus helping to maintain muscle mass and prevent sarcopenia.

Whey proteins (i.e., a soluble, fast-absorbable, high-quality milk protein) have been largely used as a test protein. In resting young volunteers, the stimulation of MPS after consumption of 10 g EAA hydrolysate from whey protein was ≈90% greater than that with casein, and ≈20% greater than that with soy [40]. Although the effect of whey protein might be short-lived [67] because of its fast absorption and the transient rise in plasma/blood amino acid concentrations, the stimulation of muscle PS following the ingestion of isonitrogenous quantities (20 g) of either casein or whey WP was sustained over 4 h in both cases, with that of WP, however, being 65% higher [68].

Ingestion of a single dose of 38 g mixed milk protein (i.e., including both fast- and slow-absorbable proteins) in young men, resulted in a time-dependent increase in postprandial muscle protein synthesis, detectable as soon as 60′ after and lasting at least up to 5 h. [69]. Also, the ingestion of 30–40 g of a vegetal protein, mycoprotein, stimulated skeletal muscle protein synthesis to an extent comparable to that of an isonitrogenous omnivorous diet [70].

**Table 1 nutrients-15-04073-t001:** Protein (and) amino acid doses (in grams, g) that increase muscle protein synthesis (MPS) in young adults.

Type of Food/Protein/AA	Subjects	Dose A: Either the Threshold or the Lowest Dose that Increased MPS	Dose B: Either the Highest Dose Tested or that Maximally Stimulated MPS	Exercise Status	Refs.
Whey protein	Y (≈21 y) males	>20 g	≈20 to 40 g	+Ex	[71]
Whey protein	Y (20–22 y) males	10 g	40 g	−/+Ex	[54]
Whey protein	Y adults	5–20 g	20 g	−Ex	[72]
Combination of 5 Whey Protein and 1 Egg studies (original data recalculated to body weight.)	Y (22 y) males	8 g	≈20 g	−/+Ex	[73]
Milk protein + CHO	Y (27 y)	15 g	45 g	+Ex	[74]
Milk protein concentrate	Y (22 y) males	/	38 g	−Ex	[69]
Egg protein	Y (22 y) males	5–10 g	20 g	+Ex	[72]
Egg protein	Y males	5–10 g	20 g	+Ex	[55]
Whey protein Hydrolysate (dose achieving the greatest stimulation of MPS in [40].) (=AA)	Y (23 y) males	≈10 g (as AA)	/	+Ex	[40]
Casein (micellar) (dose achieving an intermediate stimulation of MPS in [40].)	Y (23 y) males	≈10 g	/	+Ex	[40]
Soy Hydrolysate (=AA) (dose achieving the lowest stimulation of MPS in [40].)	Y (23 y) males	≈10 g (as AA)	/	+Ex	[40]
Beef (beef contains ≈ 22% of weight as protein.)	M males	>113 g	>170 (further enhanced by exercise) g	−/+Ex	[56]
Mixed animal (beef) and vegetal protein	Y (≈ 30 y) males	40 g	40–70 g	−/+Ex	[57]
Cristalline EAA	Y (34 y)	15 g (as EAA)		−Ex	[58]
EAA + Leucine + CHO	Y (≈26 y)	≈20 g (as EAA) (Published data are reported as 0.35 g/FFM (Free Fat Mass). Here, they have been recalculated per mean subject assuming that FFM equals LBM (Lean Body Mass).)		−Ex	[59,75]
Mycoprotein concentrate (corresponding to a total of 70 g whole mycoprotein.)	Y (21 y)	38 g		+Ex	[76]

Abbreviations used in the Table: AA: amino acid(s); CHO: carbohydrate; EAA: Essential Amino Acid(s); Ex: exercise; g: gram; M: Middle age; Y: young; y: years.

The pattern of protein administration may be important too. Diets are usually consumed as three main, bolus meals, plus occasional daily snacks. Alternatively, in some conditions and/or for experimental purposes, nutrition can be administered in a (sub)continuous pattern, i.e., at a near-constant rate over the day. In specific studies, the potential difference in the effects between these two patterns concerning the protein-anabolic effects has been investigated. Such a theoretical, experimental milieu, can be approached in vivo also by administering proteins with either a “fast” or “low” absorption pattern 

In young subjects, the total protein-anabolic effect of meal ingestion, measured using leucine tracers at the whole-body level, was more pronounced with “slow” (i.e., casein) than with “fast” (i.e., whey) protein administration [67,77]. The mechanisms leading to these effects were different, too: the “fast” protein markedly stimulated amino acid oxidation and protein synthesis but did not change proteolysis, whereas the “slow” protein increased amino acid oxidation and protein synthesis to a lesser extent but strongly inhibited proteolysis. Therefore, since the effects of whey proteins may be more rapidly vanishing, it would be useful to combine the effects of fast and slow proteins, irrespective of whether they are vegetal or animal ones. In young adults, a supplement containing a vegetal, antioxidant-rich soy protein mixed with whey protein, could prolong the improvement of the AA net balance across the leg up to ≈ 2 h post-ingestion, compared with the 20 min attained with whey alone [78].

#### 2.2.2. Effects of Exercise and Nutrition on Muscle Protein Synthesis and Accretion

Exercise is a potent stimulator of muscle protein synthesis, particularly in the recovery phase [79], and it positively interacts with protein/amino acid ingestion in the stimulation of skeletal muscle anabolism.

Most studies agree in reporting a powerful amplification by either protein or amino acid ingestion of the anabolic effects of concurrent exercise. An abundant supply of amino acids can also enhance amino acid transport and muscle protein synthesis following leg resistance exercise [80].

Muscle mass accretion following resistance exercise combined with food ingestion is observed even following an adequate habitual protein intake (≥0.8 g kg^−1^ day^−1^) [81]. A greater protein intake (1.8–3.0 g kg^−1^ day^−1^) further augments lean body mass (i.e., protein) accretion without increasing fat mass, when compared to an energy-rich, low-protein intake (≈5% of energy as protein) [81]. The high-quality whey protein, combined with resistance exercise in young adults, exerted a greater effect on MPS than equivalent doses of lower-quality proteins, such as soy protein or casein, an effect, however, still present up to 3–5 h post-exercise [55].

As reported above, a 20 g dose of whey protein was sufficient for the maximal stimulation of post-absorptive myofibrillar MPS in young men, whether or not exercising [54], and it was effective up to 3–5 h after exercise [40,82]. Conversely, others reported that the response of muscle protein synthesis following whole-body resistance exercise is greater following 40 g rather than 20 g of ingested whey protein [71]. After the ingestion of incremental doses of mixed milk protein (0, 15, 30, or 45 g) together with 45 g carbohydrate, the 30 g protein dose was sufficient to maximize the myosin synthesis rates during recovery from a single bout of endurance exercise in young men [74]. Therefore, a whey (milk) protein dose between 20 and 40 g seems sufficient for the maximal stimulation of MPS following exercise (Table 1). The greater potency of whey protein over other protein types is likely due to its nature of leucine-rich, high-quality protein, as well as to its rapid digestion and absorption, henceforth producing earlier and greater hyperaminoacidemia and hyperleucinemia [67,83].

The intake of another high-quality protein, i.e., egg protein, as low as 5–10 g (approximating that contained in a single ≈ 60 g egg: ≈ 6.8 g total protein), increased MPS above basal following resistance exercise, reaching a maximum with a dose of 20 g egg protein [72].

The structure/form of nutrient intake (free AA vs. intact protein) with respect to the timing of administration (either before, or 1 to 3 h after, exercise) on the stimulation of skeletal muscle protein accretion is relevant too. While net amino acid uptake by skeletal muscle (measured by means of femoral arteriovenous sampling) was greater when free essential amino acids plus carbohydrates were ingested before, rather than after, resistance exercise [84], the ingestion of intact whey protein was similar irrespective of the administration time [85].

#### 2.2.3. Effect of Other Substrates

Protein synthesis is an energy-requiring process [86]; therefore, energy-providing substrates such as glucose and fat may affect protein turnover too. The activities of the cellular pathways controlling protein turnover are bio-energetically expensive and therefore depend on intracellular energy availability (i.e., macronutrient intake) [87].

##### Glucose

Glucose increased muscle protein synthesis in vitro [88]. Testing glucose-induced or derived substrates separately, tissue ATP decreased during incubation with lactate, and lactate + pyruvate supported protein synthesis better than pyruvate or glucose. The data on the effects in humans of either glucose or fat on protein metabolism, specifically on skeletal muscle, are scarce, complex, and not univocal [42]. Enteral glucose administration did not affect either duodenal mucosal protein FSR or the activities of mucosal proteases [89]. An oral glucose load, and the simultaneous glucose-induced stimulation of insulin secretion, did not alter the rate of whole-body protein synthesis or breakdown [90]. Similarly, glucose ingestion added to a protein dose that maximally stimulated MPS, despite greater insulin increments, did not show either additive or synergistic effects on either the stimulation of MPS or the inhibition of muscle protein breakdown [91]. The co-ingestion of carbohydrates with protein did not further augment the post-exercise stimulation of muscle protein synthesis [92]. In humans, an acute intravenous glucose infusion decreased whole-body protein degradation, an effect possibly mediated by insulin [93]. In contrast, hyperglycemia did not inhibit whole-body protein degradation in humans under insulin-controlled conditions [94]. To our knowledge, no data are available on the direct effects of glucose per se on skeletal muscle protein synthesis in humans. The co-ingestion of carbohydrates with protein did not further increase the post-exercise stimulation of muscle protein synthesis [92]. However, the diet-induced modulation of intramuscular carbohydrate (i.e., glycogen) availability affected skeletal muscle and skeletal muscle and whole-body protein synthesis, degradation, and net balance during prolonged exercise in humans [95]. Skeletal muscle CHO stores were depleted by previous exercise in the Low-CHO group, but were replenished with a High-CHO diet for 2 days. In the Low-CHO, the net leg protein balance was the decreased group compared with both the pre-exercise rest and the High-CHO condition, primarily due to increased protein degradation and decreased protein synthesis late in exercise. Whole-body leucine oxidation increased above rest in the Low-CHO group only, and was higher than in the H-CHO group. Whole-body net protein balance was reduced in the Low-CHO group, largely due to decreased general protein synthesis. Overall, these observations suggest an effect of a High-CHO diet to improve the net protein balance in skeletal muscle undergoing exercise.

##### Lipids and Ketones

The effects of either lipid or ketone infusion/administration in humans are complex. Lipid infusion in humans did not affect proteolysis [96]. In contrast, medium-chain fatty acid infusion apparently increased leucine oxidation and, therefore, net protein catabolism [97]. Thus, the effects on whole-body protein degradation may depend on the fatty acid length [98]. The increase in FFA decreased basal muscle protein synthesis, but not the anabolic effect of leucine [99]. When associated with dietary protein ingestion, neither acute nor short-term dietary fat overload impaired skeletal MPS in overweight/obese men in the post-prandial phase, thus excluding a role by dietary accumulation of intramuscular lipids on the anabolic response to meal ingestion [100]. The infusion of 3OHButyrate decreased both whole body and forearm protein turnover (measured by phenylalanine/tyrosine tracers), as well as phenylalanine catabolism, in post-absorptive conditions, whereas it did not modify the insulin-induced effects following an euglycemic clamp [96]. In another study, 3OHButyrate infusion did not change whole-body leucine turnover, decreased leucine oxidation, and increased the non-oxidative portion of leucine disposal, while muscle PS increased, demonstrating an anabolic effect of 3OHButyrate at this level [101].

The leucine metabolite β-hydroxyl, β-methylbutyrate (HMB) was recently identified as a potentially anabolic substrate [102]. Using combined stable isotope amino acid infusion, isotope incorporation into muscle myofibrils and arterial–venous femoral sampling in humans, oral administration of 2.42 g of pure HMB increased muscle protein synthesis by ≈ 70%. In contrast, it decreased muscle protein degradation by ≈60%, in an insulin-independent manner, whereas an oral 3.42 g leucine load increased muscle protein synthesis by ≈110% [103].

##### Other Nutritional Interventions

β-alanine supplementation may increase physical performance in middle-aged individuals [104] and improve physical performance during exercise [105]. Creatine may increase muscle mass in combination with resistance exercise, although the mechanism(s) of action remain elusive. Short-term creatine monohydrate (CrM) supplementation may exert anti-catabolic actions on selected proteins in men, but it did not enhance either whole-body or mixed-muscle protein synthesis [106]. Acute metabolic studies testing various substances may provide useful information for estimating the efficacy of potentially anabolic agents [107].

### 2.3. Hormones and Related Drug Interventions

#### 2.3.1. Insulin

Although insulin has an undisputed anabolic effect on tissue protein, its experimental demonstration in vivo has challenged the investigators over time, mainly because its administration induces a decrease in amino acid plasma concentration, thus possibly obscuring its direct effect in muscle. A major advancement was the maintenance of the “amino acid “clamp” at baseline during insulin infusion or injection, also achieved when insulin was directly infused into the artery perfusing a muscle-rich organ (such as the leg or the forearm), thus avoiding a system perturbation of the aminoacidemia and allowing, at the same time, the insulin effect in the perfused limb to be studied selectively, using any of these techniques. Muscle protein synthesis and degradation were determined by combining amino acid isotope infusion with arterial–venous limb catheterization, often complemented by muscle biopsy in some studies. A systemic insulin infusion, with no prevention of hypoaminoacidemia, suppressed whole-body protein degradation [43,108], and improved limb (either the forearm or the leg) net protein balance mainly by decreasing protein degradation [43]. However, when insulin-induced hypoaminoacidemia was prevented, insulin was shown to stimulate muscle protein synthesis [109,110]. In addition, insulin enhanced the amino-acid-induced stimulation of protein synthesis [43]. In summary, the protein anabolic effect of insulin in skeletal muscle requires sufficient plasma amino acid levels; a condition that is physiologically attained following meal ingestion.

#### 2.3.2. Glucocorticoids

Cortisol, often referred to as a stress hormone, has both catabolic and anabolic effects, depending on the context. Cortisol-induced *secondary* sarcopenia (i.e., specifically induced either by either an exogenous administration or an endocrine disease) is a frequent finding in the clinical setting. Chronic glucocorticoid exposure induces loss of lean body mass by decreasing protein synthesis and increasing degradation [111,112,113]. The protein-catabolic effect of prednisone is antagonized by growth hormone [112]. In humans, the administration of 8 mg dexamethasone daily for 4 days antagonized the anti-proteolytic effect of insulin in the forearm [114]. 

#### 2.3.3. Human Growth Hormone (hGH) and IGF-1

In humans, hGH administration increased whole-body [112] and muscle protein synthesis [115]. The insulin-like growth factor-binding proteins (IGFBPs) bind to IGFs, modulating their activity and availability. They help in regulating IGFs’ actions in various tissues, including muscle. When rhIGF-I was infused at a rate achieving plasma IGF-I concentrations close to those observed following rhGH treatment, and yet avoiding the IGF-1-induced hypoglycemia, proteolysis and protein synthesis were not affected, even in the presence of prednisone treatment [116]. However, when rhGH and rhIGF-I were administered simultaneously, nitrogen balance was remarkably improved [116]. Thus, the exact anabolic effect of combined rhGH and rhIGF administration remains to be fully elucidated. Data in vitro may help to address this point.

IGF-1 exhibits several splicing variants, IGF-1Ea, which is a circulating factor synthesized in the liver, and IGF-1Eb and IGF-1Ec, recognized as Mechano-Growth Factors (MGFs), which manifest in skeletal muscles of rodents and humans, respectively [117]. The act of stretching or overloading skeletal muscles triggers a rise in IGF-1 mRNA, particularly emphasizing the specific IGF-1Ec variant (MGF) [117]. The extent to which IGF-1 alternative splicing variants might induce greater hypertrophy compared to IGF-1 per se remains partially unresolved [118]. Notably, the effects of the synthetic E-domain peptide mimetic have been elucidated: the MGF-24aa-E peptide (YQPPSTNKNTKSQRRKGSTFEEHK) activates satellite cells, prompting their replication [117]. These cells are subsequently hindered from progressing until they successfully merge with muscle fibers and assume a myogenic program [119].

#### 2.3.4. Catecholamines

Epinephrine has both an α- and β-receptor affinity. Epinephrine infusion in humans depressed plasma amino acid concentrations, particularly the essential ones, however without changing leucine or phenylalanine flux [120], nor did it impair the disposal of exogenous amino acids in humans [121]. However, in another study, the increase in plasma epinephrine concentrations inhibited proteolysis and leucine oxidation in humans via beta-adrenergic mechanism, compatible with an anabolic effect [122]. In the human forearm, physiological epinephrine exerted an anticatabolic action on muscle protein, however, with unknown mechanism(s) [123].

The β-agonist clenbuterol may exert anabolic effects in skeletal muscle [124]. However, its use in humans is hampered by unacceptable large cardiovascular side effects [125].

#### 2.3.5. Estrogens

Although estrogens are female sex hormones, they are also present in smaller amounts in males. Estrogen plays a role in maintaining bone health, promoting protein synthesis and muscle growth, and influencing body composition, in both males and females. While the exact mechanisms by which estrogens affect muscle growth are still being studied, research suggests that they can, directly and indirectly, affect muscle tissue. One way estrogens may influence muscle growth is by promoting protein synthesis, interacting with receptors in muscle cells, and activating signaling pathways involved in protein synthesis [126]. Decreased estrogen-associated signaling impairs mitochondrial function leading to muscle atrophy [127]. Estrogens can also indirectly affect muscle growth by modulating the production and activity of other hormones, such as growth hormone and insulin-like growth factor 1 (IGF-1), which are important for muscle development. Estrogens may regulate the release of these hormones from the pituitary gland and the liver, thereby influencing muscle growth [128]. Moreover, estrogens can influence muscle mass indirectly through their impact on body composition. Estrogens regulate fat distribution, and higher body fat levels have been associated with lower muscle mass. By influencing body composition, estrogens can indirectly affect muscle growth and maintenance [126]. Estrogen decreases in menopause may contribute to accelerated muscle loss and sarcopenia in females [129,130,131]. It is important to note that the role of estrogens in muscle growth is complex and can vary depending on factors such as age, sex, hormone levels, and other individual characteristics. Additionally, the research on this topic is still evolving, and different studies may have differing findings and conclusions.

#### 2.3.6. Androgens

Testosterone is a potent anabolic stimulus primarily through improvement in the re-utilization of amino acids released from protein degradation [132] (see also the above paragraph), and this will be further discussed below. Testosterone and progesterone, but not estradiol, stimulated muscle protein synthesis in postmenopausal women [133].

### 2.4. Exercise

Exercise is a powerful stimulus to promote skeletal muscle protein synthesis and net protein anabolism, involving specific metabolic and morphological adaptations in muscle [79], also interacting with substrates [80,134]. Exercise produces diverse changes in amino acid metabolism and protein turnover in muscle, according to the exercise phase. Acute changes in amino acid metabolism during exercise are primarily catabolic (i.e., increased amino acid oxidation), yet exercise does not cause muscle wasting. This is because both immediate and later post-exercise phases are anabolic. Thus, regular exercise is essential for optimizing muscle growth and hypertrophy.

The type of exercise also determines the magnitudes of these processes, and exercise requires a sequence of metabolic adjustments from the catabolic period of the ongoing exercise to the anabolic period of recovery. Two primary exercise types commonly associated with muscle growth are resistance and endurance. Resistance exercise, such as weightlifting, involves using external resistance to challenge the muscles. This type of exercise is a primary intervention used to develop strength and stimulate muscle hypertrophy. Muscle mass increases constitute key components of conditioning in the outcome of various sports due to the correlation between cross-sectional muscle area and muscle strength [135,136].

#### 2.4.1. Resistance Exercise Can Be Further Classified into Two Main Categories:

High-Intensity Resistance Exercise: This exercise type typically involves lifting heavy weights for a relatively low number of repetitions. It primarily targets fast-twitch muscle fibers, which have a higher potential for muscle growth. High-intensity resistance exercise promotes muscle hypertrophy by acutely causing mechanical tension and muscle damage, subsequently triggering muscle protein synthesis and adaptation [137]. High-intensity resistance exercise is a potent stimulus for skeletal MPS and hypertrophy in young adults during high-intensity exercise and post-exercise recovery [138,139,140,141,142,143].

Moderate-Intensity Resistance Exercise: This exercise type involves using moderate weights for more repetitions. While the hypertrophic response may be less pronounced than high-intensity resistance exercise, moderate-intensity resistance exercise still contributes to muscle growth and can benefit endurance and functional capacity [137].

Endurance exercise, or aerobic exercise, involves continuous rhythmic movements that challenge the cardiovascular system and improve endurance. Endurance exercises include running, cycling, swimming, and brisk walking. While endurance exercise primarily focuses on cardiovascular fitness and endurance, it can also impact muscle growth, particularly in prolonged training. Endurance exercise can enhance the oxidative capacity of muscles, improve energy efficiency, and increase the density of blood vessels within the muscle tissue. These changes can contribute to improved muscle function and endurance, but the hypertrophic response regarding muscle size may be relatively limited compared to resistance exercise [144].

#### 2.4.2. Effects of Exercise in Conjunction with Nutrient Intake

The specific adaptation to each type of exercise can vary depending on factors such as exercise intensity, volume, frequency, and individual characteristics. Combining resistance and endurance exercises in a well-designed training program can provide comprehensive benefits for muscle growth, strength, endurance, and overall fitness. It has been shown that combined strength and endurance training in the evening may lead to larger gains in muscle mass [145]. Additionally, other factors such as nutrition, rest, and recovery play crucial roles in maximizing the benefits of exercise and promoting muscle growth. Adequate protein intake, overall caloric balance, and appropriate recovery periods are important factors to be considered when optimizing muscle growth in response to exercise.

During the acute phase, the energy needs to stimulate amino acid oxidation/catabolism (together with that of glucose and fat), thus depleting intracellular amino acid pools. In contrast, in the recovery phase, the depleted amino acid pools and energy need to be reconstituted. Since the response of muscle protein metabolism to a resistance exercise bout lasts up to 5 h [55], such an ample post-exercise recovery phase allows a comfortable, positive anabolic interaction with food (protein) intake. Immediately following exercise, muscle protein turnover, i.e., protein synthesis, breakdown, and amino acid transport, are accelerated. However, the net protein balance remains negative (i.e., catabolic or not above zero) unless food is ingested [80]. Therefore, food intake associated with exercise is necessary to bring muscle protein balance to be positive. Nevertheless, as it is commonly experienced, it generally takes weeks to months before training-induced changes in skeletal muscle mass become apparent. The prolonged time course for hypertrophy reflects the slow turnover rate of muscle proteins, which is about 1% per day for contractile proteins [37,146].

The molecular mechanism driving the anabolic effect of exercise in muscle recognizes the mammalian target of rapamycin complex 1 (mTORC1) activation as a central, although not exclusive, mechanism regulating muscle cell size and growth [147,148,149]. The same mTORC1 mechanism stimulates human skeletal muscle protein synthesis by essential amino acids [150]. Provision of amino acids, whether in free form [150,151,152,153,154] or as intact protein [155], in association with resistance exercise, increases muscle protein synthesis and net protein accretion resulting in a positive net muscle protein balance.

While dietary protein supplementation can augment the effects of resistance exercise on the increase in skeletal muscle mass and strength, it also can preserve skeletal muscle mass during periods of diet-induced energy restriction [156]. The time relationships between nutrient intake and exercise have been the object of detailed investigations.

Both pre- and/or post-exercise nutritional interventions (i.e., by carbohydrate + protein or protein alone) effectively increase body strength and improve body composition [157]. However, the size and timing of a pre-exercise meal may also affect the delay after which the post-exercise protein feeding is optimal. The post-exercise ingestion (from immediate to 2 h after) of high-quality protein sources stimulates robust increments in MPS. A mixed meal intake immediately after resistance exercise may effectively suppress MPB (muscle protein breakdown) in the morning in young volunteers [158]

Without exercise, meal timing and administration frequency exhibited little effect on body composition and body weight, whereas meal frequency can favorably improve appetite and satiety.

#### 2.4.3. Exercise–Insulin Interaction

Insulin and exercise can positively interact in the stimulation of protein anabolism, in a complex fashion. Both protein synthesis and degradation rates are ≥1-fold greater following the post-exercise recovery than at rest [159]. The effects of insulin on muscle protein synthesis and degradation, either without or with exercise, are complex. Insulin-stimulated protein synthesis at rest, but not in the recovery post-exercise phase. In contrast, insulin decreased protein degradation following exercise as opposed to no effect at rest. Thus, the post-exercise phase can be viewed as a (transient) insulin-resistant condition as regards protein synthesis that, in terms of the effects on net protein balance, is however overwhelmed by a greater insulin-mediated suppression of proteolysis.

## 3. Part 2. The Pathophysiology of Sarcopenia in Ageing: Metabolic Hormonal, Cardiovascular, and Functional Changes in the Elderly and the Effects of Nutrition, Exercise, and Other Factors

The study of the mechanism(s) leading to ageing sarcopenia captures a fast-growing interest in both the basic and clinical–experimental sciences. While basic sciences rely on a variety of in-depth molecular, in vitro, and in vivo approaches, in vivo human studies are unique in that they translate the investigation in the human model, in turn, hampered by objective ethical/technical/investigational limitations, as outlined above.

Sarcopenia in ageing is associated with many metabolic, hormonal, cardiovascular, and functional changes. [160] The metabolic rate tends to decrease with age, resulting in a slower rate of energy expenditure, favoring weight gain and obesity with its related conditions. A reduction in muscle mass is observed, which affects mobility and overall physical function and is a major cause of diminished energy expenditure. Bone density tends to decrease with age, making older individuals more prone to fractures and osteoporosis. A decreased elasticity of blood vessels and an increased risk of hypertension and cardiovascular diseases are usually observed. Hormone levels, such as estrogen and testosterone, decrease, with effects on reproductive function, sexual health, and the gut–brain axis interplay. The immune system’s efficiency can decline, resulting in a higher susceptibility to infections and a diminished response to vaccinations. Cognitive abilities may change with age as well as vision and hearing. Digestive functions can be affected by reduced acid production by the stomach and changes in bowel motility. Older individuals may also experience changes in sleep patterns. Lung capacity and respiratory function may decrease, affecting lung health and physical endurance.

### 3.1. The Molecular Mechanisms behind Anabolic Resistance with Ageing

Anabolic resistance refers to the reduced ability of skeletal muscle tissue to respond to common anabolic stimuli, such as dietary protein and exercise, by increasing muscle protein synthesis rates (MPS). In healthy individuals aged ~18 to 50 who are not sedentary and eat sufficient daily amounts of protein and energy, skeletal muscle mass remains relatively unchanged throughout daily life [161].

#### 3.1.1. Intracellular Signaling

##### mTOR Kinase 

Basal mTOR activation is elevated during ageing and may contribute to anabolic resistance disrupting metabolic signaling pathways [162]. Markofki and colleagues aimed to investigate the effect of age on basal muscle protein synthesis and the signaling pathways involved in muscle protein synthesis, specifically the mechanistic target of the rapamycin complex 1 (mTORC1) pathway. They recruited a large cohort of young (18–30 y) and older (65–80 y) men and women to assess potential age-related differences and employed stable isotope tracer techniques and muscle biopsies to measure muscle protein synthesis rates and examine mTORC1 signaling at the basal level and in response to feeding. The study’s findings demonstrated that muscle protein synthesis rates were lower in older than young individuals. Additionally, the basal activation of mTORC1 was higher but attenuated in the older group in response to feeding. These findings suggest that age-related declines in muscle protein synthesis may be partly attributed to impairments in the mTORC1 signaling pathway. Moreover, there might be age-related differences in mTOR signaling in response to resistance exercise. Compared to their younger counterparts, older individuals experience delayed phosphorylation of mTORC1 up to 24 h after exercise and food intake, which is necessary for promoting muscle protein synthesis [162,163]. 

##### AKT Kinase 

Studies suggest that anabolic resistance in ageing muscle may involve dysregulation of Akt signaling and downstream pathways [164]. Some key points regarding Akt and anabolic resistance in older people include impaired Akt activation, blunted protein synthesis response, and insulin resistance [165]. Studies have shown a reduced Akt activation in response to anabolic stimuli, such as resistance exercise and nutrient intake, in older adults compared to younger individuals [163]. Anabolic resistance in ageing muscle is characterized by a diminished muscle protein synthesis response to anabolic stimuli, which can be attributed, at least in part, to impaired Akt signaling. Age-related insulin resistance can impair Akt signaling and anabolic resistance in skeletal muscle [166].

#### 3.1.2. Extracellular Signaling

2a. IGF-1: The role of IGF-1 in anabolic resistance, particularly in ageing and age-related muscle loss, is a topic of ongoing research. While IGF-1 is typically associated with anabolic processes and muscle growth, anabolic resistance refers to the reduced responsiveness of skeletal muscles to anabolic stimuli, leading to impaired muscle protein synthesis and loss. Ageing is associated with a decline in circulating IGF-1 levels, which can contribute to anabolic resistance. Reduced IGF-1 availability may impair the anabolic response to exercise and nutrient intake [141]. Anabolic resistance in older adults may involve dysregulation of the IGF-1 signaling pathway, including reduced activation of the IGF-1 receptor and downstream signaling molecules, such as the Akt/mTOR pathway, leading to diminished muscle protein synthesis [20]. Insulin resistance, commonly observed in older adults, can impact IGF-1 signaling and contribute to anabolic resistance [167].

3a. hGH: Several studies have explored the potential role of hGH in anabolic resistance, shedding light on its potential role in combating age-related muscle loss. One review by Velloso (2008) highlighted the interactions between hGH, insulin-like growth factor 1 (IGF-1), and muscle mass regulation, proposing that GH may play a role in anabolic resistance and the decline in muscle mass observed with ageing [168]. Older men receiving rhGH (recombinant GH) demonstrated greater muscle strength and size gains than in the placebo group. The findings suggest that rhGH supplementation in combination with resistance exercise may have potential benefits for improving muscle strength in elderly individuals [169,170]. In a study by Cuneo et al. (1991), hGH treatment was administered to elderly individuals to investigate its effects on skeletal muscle and found that GH supplementation improved muscle protein synthesis and increased lean body mass, indicating a potential role for GH in overcoming anabolic resistance in older people [171]. Finally, hGH treatment for 10 y in Growth Hormone-Deficient (GHD) adults resulted in increased lean body and muscle mass, a less atherogenic lipid profile, reduced carotid intima-media thickness, and improved psychological well-being [172]. Clinical trials investigating the efficacy of hGH for sarcopenia have produced mixed findings. While some studies have shown positive effects on muscle mass and function [173], others have yielded limited or inconsistent results. Safety concerns associated with hGH supplementation also contribute to its limited use. Potential side effects such as fluid retention, joint pain, insulin resistance, and increased risk of diseases like diabetes and cardiovascular disorders raise caution about the long-term use of hGH, particularly in older populations [174]. The cost and accessibility of recombinant hGH pose additional challenges. GH therapy is an expensive intervention, and its high cost may limit its widespread use in clinical settings, making it less feasible as a routine treatment option for preventing or curing sarcopenia [175]. In conclusion, while recombinant GH has shown potential benefits for sarcopenia in some studies, its use in clinical practice for this condition is limited. Mixed findings, safety concerns, cost considerations, and the lack of consensus guidelines contribute to the limited use of hGH in sarcopenia management. As a result, a comprehensive approach that addresses multiple aspects of sarcopenia is preferred. Future research may provide further insights into GH therapy’s potential benefits and safety profile for sarcopenia.

A study reported that serum levels of IGF-1 were increased in elderly women but not in men and correlated with frailty but not with their amount of muscle mass [176]. In elderly men, myostatin, a member of the transforming growth factor β (TGF-β) superfamily and an endogenous inhibitor of myogenesis, was increased [176]. Other studies have reported that IGF-1, myostatin, and insulin resistance were correlated with sarcopenia in elderly patients (both males and females) undergoing hemodialysis [177].

So far, it has not been precisely determined which genes are directly related to the possibility of developing sarcopenia. The studies that have been carried out so far have qualified genes predisposed to such properties that showed a correlation with muscle mass, muscle building, or impact on their functionality, to any extent. From a review that analyzed fifty-four studies on genes related to sarcopenia, twenty-six genes were identified and studied. The ACTN3, ACE, and VDR genes were the most frequently studied, although the IGF1/IGFBP3, TNFα, APOE, CNTF/R, and UCP2/3 genes were also shown to be significantly associated with muscle phenotypes in two or more studies [178].

The expression of the Mechano Growth Factor (MGF), a member of the IGF-1 (insulin-like Growth Factor 1) super family, has been shown to be both exercise and age dependent. MGF, also called IGF-1Ec, has a unique E domain with a 49bp insert in humans (52bp in rodents; IGF-1Eb), which results in a reading frame shift during the IGF-1 gene splicing to produce a distinct mature isoform. It has been administered on human cell cultures, and it has been concluded that the MGF-24aa-E peptide alone has a marked ability to enhance satellite cell activation, proliferation, and fusion for muscle repair and maintenance and could provide a new strategy to combat age-related sarcopenia without the oncogenic side effects observed for IGF1 [118]. In the context of ageing, MGF transcript production diminishes; however, when introduced to myoblast cultures derived from muscle biopsies of elderly individuals, these cells continue to exhibit replicative ability [119].

4a Anabolic Steroids: Several studies have investigated the relationship between testosterone and anabolic resistance. Bhasin et al. (2003) examined the impact of testosterone supplementation on muscle protein synthesis in older men and found that testosterone administration effectively restored muscle protein synthesis rates in older individuals, overcoming the anabolic resistance typically observed with ageing [179]. In another study, older men with low testosterone levels who received testosterone replacement therapy showed significant improvement in muscle protein synthesis rates after resistance exercise [180]. Another study investigated the effects of short-term testosterone administration on skeletal muscle protein synthesis in older men with low testosterone levels. They reported that testosterone replacement therapy improved muscle protein synthesis rates, indicating that testosterone plays a crucial role in overcoming anabolic resistance in ageing muscle; testosterone coupled with resistance exercise is an effective short-term intervention to improve muscle mass/function in older non-hypogonadal men [180,181]. Collectively, these studies may highlight the role of testosterone in combating anabolic resistance. Testosterone supplementation in individuals with low testosterone levels can restore muscle protein synthesis rates and enhance the anabolic response to exercise, counteracting the detrimental effects of anabolic resistance. Nevertheless, the use of high-testosterone doses in the elderly may be hampered by the fear of accelerating prostate cancer [182]. It is important to note that anabolic resistance is a complex phenomenon influenced by various factors, and testosterone is just one aspect of the overall picture. Other factors, such as nutritional status, physical activity levels, and other hormonal interactions, contribute to anabolic resistance. Therefore, a comprehensive approach that addresses multiple factors is necessary to manage individuals’ anabolic resistance effectively.

### 3.2. Anabolic Resistance in Ageing: Human Studies

Elderly subjects exhibit many abnormalities in protein metabolism in respect to younger subjects, here concisely summarized:An increased splanchnic “trapping” of the ingested substrates, henceforth reducing amino acid delivery to peripheral tissues, such as skeletal muscle.A decreased amino acid utilization by muscle, and/or the requirement for a greater AA load/delivery to stimulate appropriately PS in muscle, compatible with an anabolic-resistant state. In other words, the skeletal muscle in ageing might be less sensitive to lower (normal) levels of amino acids than that in young adults, and may thus require more protein to acutely stimulate muscle protein synthesis above rest, to achieve the required accretion of muscle proteins.A decrease in energy production otherwise required to sustain the energy-expensive PS.Altered protein digestion.A decrease in transluminal AA transport.An intestinal microbiota different from that of younger people.

Any of the above-listed potential factors could explain why elderly people would require, and/or are recommended to assume, at least ≈ 50% more protein than either young or mature subjects [66]. In the following sections, we report the literature data on protein turnover in ageing, both in basal and “post-absorptive” conditions, and following nutrition, exercise, and response to hormones.

#### 3.2.1. Basal Skeletal Muscle Protein Turnover in Ageing

It was initially reported that muscle protein breakdown is elevated, by as much as ≈ 50%, in the elderly compared to younger adults [183] and that muscle net protein balance is negative [184]. Similarly, a decrease in the fractional synthesis rate (FSR) of skeletal muscle proteins was initially reported in elderly subjects [142,185,186,187,188,189,190]. A −20 to 30% decrease in myosin heavy chain (MHC) FSR was observed even in middle age [191]. In respect to MHC isoform expression, that of MHC-1 did not change with age, whereas the expression of MHC-IIa isoform was decreased by ≈35% in middle age (≈54 y) as compared to young subjects, further decreasing (by ≈50%) at older ages [191]. A ≈ 50% reduction in mitochondrial protein FSR was also demonstrated in middle age, not further decreasing at ≈75 y. In addition, a decrease in the activities of mitochondrial enzymes was observed in muscle homogenates from aged people, consistent with a decreased muscle oxidative capacity [186]. Mitochondrial ATP production measured using various substrate combinations was also lower in the elderly than in young subjects [192]. Moreover, genes and proteins related to mitochondrial shaping proteins decreased in sedentary elderly [35]. Pooled data from ≈150 healthy subjects of both sexes aged 18–89 y demonstrated a progressive decrease in mitochondrial DNA and mRNA abundance and mitochondrial ATP production with advancing age [190].

However, at variance with these earlier reports, more recent studies consistently failed to confirm the earlier findings of decreased basal muscle protein synthesis in elderly subjects, showing little or no differences between young and old adults [60,138,193,194,195,196]. These opposite, contrasting results could have been due to several previously unaccounted factors, such as the experimental methods themselves, the sample populations studied (particularly regarding the general health status, diet, habitual physical activity, or else), or other unappreciated causes, underlying the complexity, as well as the limitations, of the in vivo investigations. On the other hand, should basal daily muscle protein synthesis rates in the aged subjects be lower (by ≈20 to 30% or more), than that of younger people, as originally reported, these figures would end up in a more marked, unrealistic muscle wasting than what is typically observed (estimated as 0.5–1.5% per year between 50 and 80 y old subjects, or 3–8% per decade), as well as into complete muscle loss [197], or even into negative numbers, when projected over years.

It should, however, be recognized that basal muscle protein turnover could particularly be altered in frail, elderly subjects, being potentially associated also with a chronic, subtle, systemic inflammatory state and/or other co-morbidities [198,199]. Indeed, inflammatory mediators, such as cytokines, particularly TNFα, may impair skeletal muscle protein FSR, by interfering with the phosphorylation of the mammalian target of the rapamycin (mTOR) pathway [150], critically involved in the regulation of mRNA translation, muscle protein synthesis, and growth [200]. Therefore, it is currently accepted that basal skeletal muscle protein turnover is near-normal in healthy elderly subjects. In contrast, a different, complex picture can emerge when studying the response in ageing of skeletal muscle protein turnover to anabolic stimuli, such as substrates (i.e., mixed meals, protein, amino acid mixtures, other metabolites), exercise, anabolic hormones, or their combinations.

#### 3.2.2. Skeletal Muscle Protein Turnover in Ageing in Response to Nutrition and Exercise

Elderly subjects exhibit some peculiarities in the handling of oral feeding. Using the essential amino acid leucine as a tracer, a greater first-pass splanchnic uptake of ingested amino acid(s) was reported in the elderly rather than in young people, suggesting that a lower proportion of ingested amino acid reached the peripheral circulation [195]. Such a greater splanchnic extraction could limit the amino-acid-mediated stimulation of muscle protein synthesis in peripheral tissues such as skeletal muscle. Consequently, sustaining splanchnic vs. muscle protein synthesis could indicate a “metabolic priority” during recovery from metabolic stress in healthy elderly persons. Such a mechanism might become more relevant in older individuals suffering from chronic diseases and/or subjected to poly-medications [201]. However, in contrast with the above report, it was also reported that oral amino acids stimulated muscle protein anabolism in the elderly, despite the higher first-pass splanchnic extraction [202].

A protein pulse rather than a spread (or continued) oral feeding stimulated the best protein accretion in the elderly [203], but not in young women, in whom the protein feeding pattern did not affect protein accretion. In older men, rapidly absorbed whey protein stimulated postprandial muscle protein accretion more efficiently than the “slow protein” casein hydrolysate [82]. The whey protein effect was superior to that of casein hydrolysate too, likely due to the combination of the whey protein’s fast digestion and absorption with its greater leucine content. Conversely, in younger subjects, a slowly digested protein (casein) achieves a better anabolic effect than a rapidly digested one [77]. Thus, fast-absorbable protein (or protein supplements), rich in well-balanced EAA, could be ideal and specific in the nutrition of aged people [204,205]. The cooking mode is also important. Well-done, more-digestible meat is assimilated better than rare meat in the elderly [206].

Other protein types, such as the vegetal wheat protein (administered in a 35 g dose) increased muscle protein synthesis rates in healthy older men, which was nearly as much as 35 g whey protein hydrolysate over 2–4 h. [207].

Generally speaking, dietary protein supplementation can augment the gains in skeletal muscle mass and strength mediated by resistance exercise, and can preserve skeletal muscle mass during periods of diet-induced energy restriction [208]. A bout of aerobic exercise increased the anabolic effect of nutrient intake in older adults **[209]**. Such an effect could have been driven, in addition to the protein themselves, through an exercise-induced augmentation of nutrient-stimulated vasodilation, resulting in increased nutrient delivery to muscle, rather than through improved insulin signaling. Conversely, others did not show any additional effects of protein over that induced by exercise itself, on skeletal muscle hypertrophy. Verdijk et al. compared the increment(s) in skeletal muscle mass and strength following 3 months of resistance exercise training, with or without protein ingestion, either prior to or immediately after each exercise session in elderly males who habitually consumed about 1.0 g protein/kg per day [210]. They concluded that timed protein supplementation prior to and after each exercise bout did not further increase skeletal muscle hypertrophy.

#### 3.2.3. Anabolic Response in Skeletal Muscles of Aged People

The studies aiming at detecting the existence of resistance to anabolic stimuli in ageing (i.e., of “anabolic resistance”) are complex and sometimes inconclusive, often due to the variability of the experimental setting. Several factors are to be considered, such as the amount, the quality, and the administration pattern of the protein, the energy content of the meal, etc. The same considerations should apply to the effects of amino acid mixtures. Furthermore, when exercise is combined with nutrition, the anabolic response can vary with the exercise intensity, whether it is applied to the whole body or to a single leg, with the timing of food administration, etc. Therefore, any conclusion drawn from individual studies should be balanced considering many concurrent variables. A summary of available data is reported in Table 2.

Data consistent with a normal anabolic response (= no resistance).

Several studies reported a normal response of MPS to oral administration of either protein or free amino acids in the elderly. In a dose–response study performed by feeding frequent small boluses of liquid meals to healthy > 60 y old subjects, Welle and Thornton [211] reported that a comparable stimulation of MPS was attained independently from both the protein load and the protein contribution to total energy, i.e., from low (7% corresponding to 0.6 g protein/kg/day) to moderate (14% total energy, i.e., 1.2 g protein/kg/day) or high (28% total energy, i.e., 2.4 g protein /kg/day). They concluded that in older subjects, there was no dose-dependent effect of the amount of protein intake on MPS; therefore, the “lower” protein dose was sufficient to stimulate MPS maximally. The lower dose tested in this study was comparable to that found in young subjects of other studies (Table 2).

Similarly, the administration of an EAA drink (with an amino acid composition approximating their distribution in muscle protein) to both young and elderly subjects stimulated muscle protein synthesis to the same extent in both groups [58], although the response in the elderly was retarded albeit still sustained, as opposed to the faster, short-lived one of the young.

No impairment of muscle protein synthesis to protein intake was detected in elderly subjects after ingestion of large amounts of carbohydrates and proteins [211], or of either moderate (≈115 g) [212] or large (≈300 g) amounts of beef [213]. By comparing sarcopenic (≈80 y) and healthy (≈70 y) older men, the ingestion of ≈20 g of a leucine-enriched whey protein load, increased muscle protein synthesis rates to the same extent in both groups [214].

In older adults, following the ingestion of mixed meals containing combinations of animal (beef) and vegetal proteins, the whole-body anabolic response linearly increased with increasing protein intake, primarily due to the suppression of protein breakdown. Notably, muscle protein synthesis (i.e., one factor contributing to the net protein accretion, or balance, together with the suppression of protein degradation) was further stimulated by a protein dose (70 g) above that previously considered as “optimal” in the elderly (≈35 g/kg BW) [215], yet attaining the same maximal response as that of young subjects. However, at the 40 g dose, the stimulation of muscle protein synthesis in the elderly was lower than that observed in the young volunteers, therefore compatible with anabolic resistance in the former group at a “lower” (i.e., 40 g) mixed protein dose (see also below). The above-reported protein dose(s) stimulating MPS in older subjects were, however, greater than the 20 g high-quality, whey protein dose that produced the maximal effect on MPS in young people, either exercising or not [72]. The muscle protein synthetic response to the combined ingestion of protein and carbohydrate (i.e., an additional energy source) was not impaired in healthy older men [216]. Similarly, the co-ingestion of carbohydrates with protein and free leucine stimulated muscle protein synthesis to the same extent in young and elderly lean men [217].

**Table 2 nutrients-15-04073-t002:** Effects of protein-rich food ingestion on Muscle Protein Synthesis (MPS). According to the study type, the data are grouped as Dose–response studies in the elderly (*n* = 1 study). Sarcopenic vs. healthy elderly subjects (*n* = 1 study). Studies reporting similar responses in old and young subjects (=no anabolic resistance) (*n* = 7 studies). Studies reported a decreased/blunted/delayed response (=anabolic resistance) in old subjects compared to young controls (*n* = 10 studies).

Type of Food/Protein Tested	Subjects	Dose A: Either the Threshold or the Lowest Dose Increasing MPS	Dose B: Either the Highest Dose Tested or that Maximally Stimulating MPS	Exercise Status	Comment	Reference
***Dose***–***response study***						
Liquid meals (recalculated from original data to weight and total protein intake over the test)	Healthy (62–75 y) males and females	29 g	115 g	−/+Ex	Max effect obtained at the lowest dose. MPS increases greater after exercise.	[211]
** *Sarcopenic vs. healthy elderly* **						
Leucine-enriched whey protein	Sarcopenic (81 y) males	21 g	/	−Ex	Similar increase in MPS in both groups	[214]
	Healthy (69 y) males					
** *Similar response to controls (= no resistance)* **						
Whey protein isolate	Old (71 y) males	10 g	40 g	−/+Ex	Exercise enhanced max effect at 40 g protein	[218]
Beef (beef contains −22% of weight as protein)	O (68–70 y) vs. Y (34–41 y)	113 g	340 g	−Ex		[212,213]
CHO + WP + Leu (Data recalculated for the subjects’ average weight (75 kg). The table reports the total of the EAA+Leu administered, which were fractionated in 6 doses every hour)	O (75 y) vs. Y (20 y) males	/	72 g WP + 13.5 g Leu	+Ex	30 min of moderate-intensity physical activity	[213]
EAA drink	Elderly (67 y) males and females	15 g	/	−Ex	Retarded albeit still sustained response in the elderly; faster, short-lived one in the young over 3 h	[58]
Mixed animal (beef) and vegetal foods	Elderly (69 y) males and females	35 g protein	70 g protein	−Ex	≈5x greater response of MPS at the higher protein dose. At the 35 g dose, the response in O was > 5x lower than that in Y (see below)	[216]
Mixed animal (beef) and vegetal foods	Y (31 y) males and females	40–44 g protein	66–70 g prot	−/+Ex	At the 36 g dose, the response in O was > 5x less than that in Y	[57]
Protein and CHO	O (75 y) males	20 g		−Ex		[217]
	Y (21 y) males					
** *Decreased/Blunted/Delayed response (= anabolic resistance)* **						
Combined analysis of Whey Protein (*n* = 5 studies) Egg (*n* = 1 study) (original data recalculated to body weight)	Elderly (≈71 y) males	8 g	≈32 g	−Ex	Delayed response in the older group	[73]
	Young (≈22 y) males	8 g	≈20g	−Ex		
Intact whey protein	O (≥70 y)	5–20 g	≥20 to 40 g	−/+Ex	In the resting elderly, the response of MPS plateaus at 20 g, at a lower value than in Y (40 g). Resistance exercise can increase MPS only in O but at greater protein intake.	[72,218]
	Y (> 23 y)	5–20 g	20 g	−/+Ex		
Crystalline EAA	Y (28) vs. O (70 y) (Similar response of MPS in the two groups)	2.5 g (?)	20–40 g (the highest dose (40 g) was tested only in the older group)	−/+Ex		[60]
AA + Leucine	O (68–70 y) males and females	15 (Published data are reported as 0.35 g/FFM (Free Fat Mass). Here, they have been recalculated per mean subject assuming the FFM individuals’ LBM (Lean Body Mass).)	/	−/+Ex −/+insuin	MPD = pre and post ex, with or without insulin	[59]
Leucine-enriched EAA	O (67) vs. Y (29)	6.7 g EAA	/	−Ex	A higher leucine dose (41%) was necessary to match the increase in MPS of the O to that of the Y	[195]

Abbreviations used in the Table: EAA: Essential Amino Acid(s); Ex: Exercise; O: Old; Y: Young; y: years; WP: Whey protein.

Data consistent with an impaired response (=anabolic resistance)

Other studies are, however, compatible with the existence of some degree(s) of anabolic resistance in the skeletal muscle of the elderly. Resistance in the elderly can be demonstrated following the administration of either protein or amino acid, as well as after hormone, exercise, or their combinations.

In the elderly, the dose of high-quality dietary protein administered as a single bolus (0.40 g protein/kg BW/meal, corresponding to ≈ 32 g protein/meal in an 80 kg person), required to stimulate myofibrillar protein synthesis maximally, was greater than that required in young controls, (0.24 g protein) [73].

When crystalline EAA were administered, the literature data were somehow contrasting. It was initially reported that 2.5 g crystalline EAA were sufficient to elicit an increase above basal of MPS in both young and elderly subjects, whereas resistance of MPS in the elderly was apparent only after 10–20 g EAAs [60]. However, in subsequent studies using intact whey protein, it was shown that the lower dose capable of eliciting a response of MPS in the elderly is greater than that in the young, and that the ageing muscle responds maximally at 20 g EAAs, a dose ≈ 2× lower than that in the young (40 g EAAs) [60]. In the elderly, however, MPS could be further increased with a 35–40 g intact whey protein intake. Thus, in the elderly, the dose of high-quality whey protein required to attain a maximal stimulation of skeletal muscle protein synthesis was about onefold greater than that of younger subjects [72].

A ten-day administration of a supplement containing fast-digestive proteins (soluble milk proteins compared to casein alone) could overcome muscle anabolic resistance in the elderly [205]. Furthermore, oral EAA ingestion exhibited a retarded albeit still sustained response of MPS in the elderly as compared to young subjects [58].

In older men, skeletal muscle was also less responsive to the anabolic effects of leucine in the postprandial phase than in the young controls [219]. Also, a higher proportion of leucine within an essential amino acid mixture was required for the optimal stimulation of muscle protein synthesis in the elderly, rather than in young subjects [195].

#### 3.2.4. Insulin Resistance

Insulin resistance on muscle protein synthesis in ageing was reported in another study showing that leg phenylalanine net balance, although not different between young and old subjects at baseline, was significantly increased in both groups with hyperinsulinemia, but to a greater extent in the young [220].

Nevertheless, increasing insulin availability through a local insulin infusion increased amino acid uptake but did not enhance muscle protein synthesis rates in healthy young and older men following the administration of a casein-based protein drink [221]. Thus, in the post-prandial phase, skeletal muscle appears to be in an insulin-resistant state to the stimulation of protein synthesis, without difference between young and elderly subjects (see also above).

#### 3.2.5. Resistance to the Anabolic Effects of Exercise

Resistance to exercise-induced protein anabolic effects could be another factor favoring sarcopenia. Multiple pieces of evidence exist supporting this view. Following a single bout of resistance exercise in the post-absorptive state, the dose–response curve of the stimulation of protein synthesis by resistance exercise intensity reached the maximum at 60–90% 1 RM, but was shifted to the right/low and blunted in the elderly compared to young controls, up to 1–4 h post-exercise [138]. At the molecular level, the phosphorylation of p70s6K and 4EBP1 at 60–90% 1 RM was decreased in the older group. Phosphorylation of p70s6K 1h post-exercise at 60–90% 1 RM predicted the rate of MPS at 1–2 h post-exercise in the young but not in the old. These data indicate that older men exhibit resistance to anabolic exercise of both MPS and intracellular mediators. In another study, although the acute MPS response to combined resistance exercise and EAA ingestion was comparable in young and older men, such a response was delayed with ageing, and it was associated with unresponsiveness of ERK1/2 signaling and of AMPK activation [222]. Similarly, in post-absorptive elderly subjects, following a single bout of resistance exercise, MPS increased to a lesser extent in the recovery phase than that observed in young controls [163].

At variance with the above data, however, other reports show that aerobic exercise could overcome age-related (insulin) resistance of muscle protein anabolism following a leucine-enriched essential amino acid–carbohydrate mixture, through an improvement of the endothelial function and of the Akt/mammalian target of rapamycin signaling [223]. Thus, exercise combined with EAA administration could effectively counteract age-associated sarcopenia and other conditions leading to muscle wasting. Conversely, no additional effect of heavy resistance exercise on myofibrillar protein synthesis was detected in elderly men after milk protein and carbohydrate ingestion [224].

#### 3.2.6. Deleterious Effects of Bed Rest in the Elderly

A condition opposite to exercise is bed rest, an undesired situation very common in hospitalized older subjects, as well as in those old persons in whom, for a variety of factors, physical activity is restrained and/or abolished, either voluntarily or not [156,225,226,227]. In older adults, bed rest significantly restrained the EAA-induced increase in MPS, through a reduction in mTORC1 signaling and amino acid transport [228]. EAA supplementation above RDA may help to preserve muscle function in the elderly during inactivity [229].

#### 3.2.7. Effect of the Inflammatory State

Acute and chronic inflammation retain undesired effects on skeletal muscle mass and protein metabolism. Thus, age-associated inflammation (either subtle or overt) may negatively affect the anabolic sensitivity of skeletal muscle in the elderly. Toth et al. reported a strong relationship exists between MPS and circulating concentrations of several markers of immune activation [199]. At the mechanistic level, cytokines, in particularly, TNF-α, may impair MPS by blunting the phosphorylation of proteins in the mammalian target of the rapamycin (mTOR) intracellular signaling pathway [200].

#### 3.2.8. Role of Blood Perfusion of Skeletal Muscle

Effective blood perfusion is another variable conditioning the delivery of anabolic substrates and hormones to muscle. Basal leg blood flow was greater in young than in elderly subjects, and it was stimulated in response to physiologic hyperinsulinemia in the young but not in the elderly [220]. Furthermore, a positive relationship was found between the insulin-induced changes in blood flow and the increase in muscle protein synthesis. Conversely, the stimulation of muscle protein FSR responded similarly to exogenous amino acids in healthy younger and older adults under conditions of comparable NO-induced hyperemia, and the maintenance of an adequate blood flow perfusion may have a favorable impact on protein synthesis in ageing skeletal muscle [230].

### 3.3. Strategies to Counteract Anabolic Resistance in Ageing

A number of strategies can be applied in the effort to optimize muscle protein anabolism and/or to combat anabolic resistance in the elderly. They can exploit the effects of nutrition, exercise, or both, as well as that of other specific interventions.

#### 3.3.1. The Effect of Complex Nutritional Supplements

Although natural foods rich in high-quality protein (dairy products, meat, and egg) can adequately stimulate protein anabolism as efficiently as that of protein-rich supplements and/or of specifically designed amino acid mixtures, the advantage of using specific nutritional supplements remains questionable [231]. In older women, the intake of 3 g of a leucine-enriched EAA supplement (LEAA) (containing 1.2 g leucine) stimulated MPS as much as 20 g WP (containing 2 g leucine). Thus, the composition of EAA (rich in leucine) rather than the AA/protein amount seems to be crucial to stimulating skeletal muscle anabolism [232]. When comparing the effects of a soy and milk blend with whey protein alone, on post-exercise muscle protein synthesis, the anabolic effects were similar in older subjects [233], at variance with what was observed in the young, in whom the soy and milk blend helped to prolong the stimulatory effect on MPS from 2 h (with whey protein alone) to 4 h. The co-ingestion of carbohydrate and fat with an isonitrogenous nutritional supplement (21 g of leucine-enriched whey protein) did not affect nor further improve postprandial muscle protein synthesis in older men, showing that the provision of extra energy does not modulate muscle protein synthesis [234]. The ingestion of casein added to a milk matrix, modulated dietary protein digestion and absorption kinetics, however, without affecting postprandial muscle protein synthesis in older men [208].

In a clinical trial of a 3-month supplementation period with 15 g EAA vs. placebo in older women (≈70 y), the acute anabolic response of skeletal muscle FSR to EAA supplementation was maintained over the observation period; however, LBM increased only in the supplemented ones but not in the controls, whereas muscle strength was unchanged in both groups [235]. Thus, EAA supplementation may help in improving LBM and combat, with yet uncertain mechanism(s), the debilitating effects of sarcopenia. In another study, EAA supplementation with exercise in sarcopenic older women improved speed gait but not lean mass or strength. In elderly subjects supplemented with 15 g EAAs three times daily for ten days of bed rest, functional parameters were improved following inactivity [229].

The supplementation of either protein (whey and soy), leucine, or creatine did not improve the training-induced adaptations in pre-frail and frail elderly, regardless of sex. Therefore, these findings would not support using supplements to expand the effects of resistance exercise to counteract frailty-related muscle wasting dynapenia [236].

At variance with some of the above-listed reports, even at low EAA ingestion, healthy older adults trained for 12 w with progressive bouts of resistance exercise, exhibited a normal increase in muscle strength, cross-sectional area, and mixed muscle protein FSR [237], via the stimulation of the mTORC1 complex, thus showing that a healthy, exercise-conditioned condition may help to maintain normal sensitivity to the anabolic effects of EAA and muscle protein accretion.

On the whole, many variables may account for the inter-study variability and the end-point message(s), which could appear somewhat confusing from the reported studies. This might be due to the specific outcomes selected, the doses and the duration of EAA supplementation, or the general, health-related conditions of the enrolled subjects, habitual physical activity, etc. Also, some of the chosen endpoints (such as the increase in MPS) may not really reflect a true improvement in muscle function.

In addition, it should be considered that the administration of nutritional supplements in elderly subjects at nutritional risk may be offset by a simultaneous reduction in voluntary food intake [238,239].

#### 3.3.2. Specific Effects of Leucine Addition

The addition of leucine to other nutritional products may be a valid supplement to be used in the elderly, although with somehow limited effects [240]. Older subjects consuming a leucine supplement showed a greater increase in MPS rates from baseline than the controls, but not in lean body mass or muscle function [61]. In a recent study in older men (74 y), co-ingestion of 2.5 g leucine with 20 g casein resulted in a 22% higher muscle protein synthetic rate compared with ingestion of casein alone [241].

The leucine effect seems to be dose-dependent. Leucine added as a supplement (3–5 g) to either whey protein or EAA solutions in either younger or older men, showed comparable increments in skeletal muscle myofibrillar protein synthesis (MyoPS), at variance with no sustained stimulation observed in control subjects receiving only 1.8 g leucine [242,243]. However, the leucine effect in skeletal muscle of aged people might be impaired, as mTORC1 activation is defective, and sensitivity and responsiveness of muscle protein synthesis to amino acids decreased [50]. Conversely, the basal activation of mTOR seems to be higher. A combined effect of age-related impairment of muscle signaling and insufficient availability/delivery of nutrients and growth factors to the muscle might contribute to sarcopenia. Thus, whether ageing per se affects mTORC1 signaling is yet uncertain, because of the common association between poor protein assumption, reduced/absent physical activity, and concurrent diseases. In studies in which habitual protein intake exceeded 1.0–1.1 g/kg/day, and included a moderate/high proportion of dairy protein, prolonged leucine supplementation (2.5 g daily for six months) did not increase muscle mass or strength in healthy type 2 diabetic older adults [244].

#### 3.3.3. Exercise Strategies

The primary target(s) of sarcopenia prevention/amelioration is the reduction in the risk of falls and fractures and the maintenance of independence in everyday tasks. The interventions applied to attain these targets look very similar to those aimed at the amelioration of general health status with ageing, such as improvements in muscle mass and strength, bone density, cardiovascular fitness, maximal oxygen consumption, endurance, and energy metabolism, in addition to the reduction in insulin resistance and the associated risk of diabetes mellitus, coronary heart disease, hypertension, and obesity [245]. In other words, the benefits of regular exercise translate into an improvement in the quality of life of elderly populations. Practicing regular exercise is a prerequisite to attain and maintain these benefits over time. In order to attain these key objectives, much attention has to be devoted to the implementation of specific exercise programs, aiming at increasing the awareness of their safety, required constancy, and compliance.

Following regular training, older subjects can increase muscle strength as much as that of younger control subjects, even by threefold, over a few months. This result is initially accomplished by neural adaptation(s) and greater muscle fiber recruitment, while more prolonged resistance training leads to an increase in muscle mass/size too, (partially) restoring the loss of the cross-sectional area of type II muscle fibers occurring with ageing [246]. Resistance training contributes to improvement in the functional capacity of the levels of physical activity and allows participation in daily living activities to be maintained, thus being motivationally efficient too. Specific protocols for the implementation of exercise to combat sarcopenia in ageing, in the old [247], as well as the oldest-old (80–85 y) [248] and the physically frail [249], have been proposed.

Both resistance/strength/weight exercise and endurance/aerobic training of skeletal muscles may be useful in the prevention and treatment of sarcopenia. Strenght training can positively affect the neuromuscular system, and increase hormone concentrations and the MPS rate [250]; however, from a recent meta-analysis, the benefit of combining dietary supplements and exercise is not so straightforward among different populations [251].

Exercise training has been consistently shown to be highly effective in slowing down and possibly preventing the insurgence of sarcopenia in older people, counteracting the loss of muscle mass, and strength, and an increase in intramuscular fat infiltrations. While stressing the importance of a personalized approach, general physical activity guidelines suggest that a combination of aerobic exercise, progressive resistance training, and balance training might be the ideal intervention for a sarcopenic population. Of note in a mixed training involving aerobic, strength, and balance exercises was effective in improving or preserving motoneuronal health and neuromuscular junction (NMJ) stability, together with muscle mass, strength, and functionality in an old, sarcopenic population [252]. These sarcopenic subjects were trained three times per week for 2 years with a mix of aerobic, strength, and balance exercises matched with nutritional advice [252]. The same group of researchers followed older dancers for 6 months, reporting a major stability in neuromuscular junctions and a superior functional performance despite no differences in muscle size [253].

#### 3.3.4. Steps for Adaptation to Exercise

First, a physical assessment of the functional status of the elderly is mandatory. Elderly adults are commonly more susceptible to orthopedic traumas and cardiovascular complications. Therefore, an accurate evaluation of their actual independence with respect to daily life activities (ADL), and of cardiovascular tolerance and conditions, should be performed. Patients with already compromised ADLs should be tested more specifically for sarcopenia. The evaluation of the risk of falls, and their prevention, should be part of the initial treatment strategy. In the case of resistance training programs, the elderly should be tested for their CV fitness as well as properly instructed on safe lifting techniques.

A proper warm-up step should precede the onset of resistance training. Since a loss in elasticity of muscle and connective tissue, and the subsequent increase in stiffness, are common with advancing age, so a preliminary step would reduce the risk of injuries in older subjects. Resistance exercise should start at a low intensity, and then gradually increase over time. Strength training should begin with relatively light weights, being comfortably lifted through a full range of joint rotation/motion while maintaining a proper posture and mechanics. Individuals should avoid holding and/or deeply changing their breath frequency during force production (i.e., perform a Valsalva maneuver). The lifted weight should be gradually increased paralleling the improvement in strength, with the goal to meet the chosen, relatively high-intensity resistance training program, which has proven to be the most effective in older populations [254].

The exercise intensity is usually targeted to a percentage of the individual’s one-repetition maximum (1RM is the maximum weight that can be safely lifted one time). Exercise intensity is calibrated to a weight allowing one set of 8 to 15 repetitions to be completed leading to volitional fatigue (i.e., between 8- and 15RM), approximately corresponding to 60 to 80% of 1RM. This intensity leads to significant improvements in muscle strength and mass. [255]. Although higher exercise intensities (up to 85 to 100% of 1RM) may be associated with greater improvements in strength, they may increase the risk of musculoskeletal injuries in older subjects. When an increase in muscle endurance is pursued, a lower-intensity weight that can be lifted for more repetitions is recommended. [245].

The duration (i.e., the volume) of resistance training necessary to produce the maximum benefits in muscle strength is currently under debate. Usually, multiple sets (2 to 3 sets) of 8- to 15RM are recommended. Conversely, others suggest that a single set is as efficient as multiple sets for strength gains in adults. [255]. Notably, increased compliance to a single set of exercises may allow the completion of the resistance training program in a shorter period of time, thus reducing the rate of dropouts. [256,257]. Albeit a single set seems to be sufficient to produce significant gains in strength, multiple sets can be employed when greater strength gains are desired.

Improvements in muscle strength are usually observed with a training frequency of 2 to 3 days/week in the elderly. While increasing the frequency to 4 to 5 days/week may induce further strength gains, a decrease in compliance may however ensue. Thus, a frequency of 2 to 3 days/week would be recommended for elderly individuals beginning a resistance training program [245].

Ideally, resistance training is directed at the large muscle groups of the arms/shoulders, chest, back, hips, and legs that support everyday activities. Each repetition should be performed slowly through the full option of motions, with sufficient time to lift the weight (i.e., concentric exercise) as well as to lower the weight (i.e., eccentric exercise). Thus, appropriate, variable-resistance machines using weight stacks should be employed in this population, as they reduce the risk of injuries to extremities, and decrease the risk of lower back injuries and that of exercised-induced hypertensive spikes, because the weights can be progressively titrated in small increments, and resistance can be applied through an ample range of motion. [245].

#### 3.3.5. Caution and Contraindications to Exercise in the Elderly

While healthy older adults should safely engage in some mild to moderate physical activity, caution should be taken about the planning of more vigorous exercise in these subjects. The American College of Cardiology and American Heart Association recommends a test of exercise stress in yet asymptomatic men older than 40 years, as well as in women older than 50 years, before the initiation of vigorous exercise programs [258,259,260].

Specifically, in older adults with existing disease, the risks associated with regular exercise may outweigh its potential benefits. As a matter of fact, exercise may elicit/aggravate pre-existing diseases and conditions common to older adults, such as arthritis, osteoporosis, angina, and hypertension at advanced stages. An accurate collection of their medical history, a thorough physical examination and risk stratification, and a close supervision and education performed by trained personnel can minimize the potential risks and hazards of exercise in the elderly [258,259,260,261].

#### 3.3.6. Other Treatments

Although the positive effects of testosterone replacement in elderly (hypogonadal) men have been outlined above, only modest increments in muscle mass and strength were produced, and not observed in all studies [182]. The potential for an acceleration of prostate cancer should be taken into account too. Similarly, the replacement of hGH in elderly subjects did not increase muscle strength and did not expand strength gains following resistance training [182]. The pharmacological stimulation of the GHRH-IGF-I axis including its major circulating binding protein (IGF-I/IGFBP-3) is yet to be fully tested and investigated. The potential of myostatin inhibition in the treatment of sarcopenia associated with chronic kidney disease is under consideration [262][]. New exercise technologies, and supplementation by antioxidants, vitamin D, eicosapentaenoic acid, or ursolic acid, as well as activation of peroxisome proliferator-activated receptor γ coactivator-1α (PGC-1α), may be considered too [263].

#### 3.3.7. Optimizing Nutrition–Exercise Interaction in the Stimulation of Skeletal Muscle Anabolism in Ageing

The magnitude of the anabolic response to nutrients combined with exercise may be influenced by factors other than just the amount of a nutrient ingested or the magnitude of exercise. The daily distribution and timing of protein ingestion, the co-ingestion of different nutrients, and the type/quality of the protein or of the amino acid mixture ingested may all influence protein accretion (Table 3).

A number of reports have also investigated the optimal daily distribution of food protein, as well as the timing of the administration with respect to exercise performance (see also above).

In regard to food protein distribution over the three main daily meals, it was reported that non-frail elderly subjects consume dietary proteins more evenly than pre-frail and frail subjects, thus suggesting that the daily protein load should be consumed as ≈30 g protein (in a representative 75 kg subject) at each of the three daily mealtimes [264]. Conversely, in a subsequent randomized-controlled trial, the pattern of meal-protein intake (i.e., even vs. not even) over the day did not affect lean body mass, muscle strength, or other functional outcomes, as well as whole-body protein kinetics and MPS over 8 weeks in older adults [265].

With respect to the timing of food protein food administration with exercise (i.e., before, at the beginning, i.e., at t = 0′, or after exercise performance), in the only published study in the elderly, the early intake of a protein supplement immediately before (at t = 0′) each bout of resistance-type exercise in a 12-week intervention study in the elderly sustained skeletal muscle hypertrophy, as opposed to having no effect with the supplement intake taken 2 h after exercise [266]. Notably, protein ingestion in the evening/night before sleep increased Muscle Protein Synthesis throughout the night in healthy older men [267].

A list and the quantities of common foods providing ≈ 30 g of protein are reported in Table 4. (data compiled from ref. [268]) The quantities of animal foods are lower than those of vegetal foods (except for soybeans and quinoa), because of their balanced content of essential amino acids. The reported quantities are only indicative, because most of these foods have not been specifically tested in controlled studies of stimulation of skeletal MPS with exercise in the elderly.

Besides protein or amino acid supplements, other substances could be considered to promote skeletal muscle accretion [269].

#### 3.3.8. Muscle and Bones

The idea of a bone–muscle unit has emerged in the past decades. Different studies have provided valuable insights into the intricate communication between muscle and bone during anabolic resistance [270,271,272]. These investigations have revealed that muscle and bone are tightly interconnected tissues, with reciprocal interactions and shared regulatory mechanisms. Mechanistically, it has been established that mechanical loading, generated by muscle contractions during physical exercise, plays a pivotal role in preserving bone health. By applying forces to the bone, mechanical loading stimulates bone remodeling and fosters the maintenance of bone density and strength [273]. Conversely, disrupting this mechanical loading during periods of muscle disuse, such as immobilization or inactivity, can result in bone loss and increased susceptibility to osteoporosis [274]. Moreover, molecular mediators released by muscle cells, including growth factors, hormones, and cytokines, have emerged as crucial signaling molecules facilitating the bidirectional crosstalk between muscle and bone. These molecular messengers transmit signals and exert regulatory effects on bone cell activity, ultimately influencing bone turnover and remodeling processes. From the muscle can be released osteoinducer (IGF-1, FGF-2, IL-15, OGN, FAM5C, Tmem119, and osteoactivin) and osteoinhibitor (IL-6 and myostatin), among different stimuli [275]. Consequently, interventions targeting anabolic resistance and aiming to preserve or enhance muscle mass and function, such as resistance exercise and optimizing protein intake, hold considerable potential to positively impact both muscle and bone health [276]. A comprehensive understanding of the intricate communication between muscle and bone during anabolic resistance is of paramount importance for the development of effective strategies to combat age-related muscle and bone disorders.

## 4. Conclusions

Recent research has produced a huge amount of data about the molecular and nutritional, as well as lifestyle- and health-related factors that can affect the development and the worsening of sarcopenia in the elderly. These data could well be exploited for the design of interventions aimed at optimizing skeletal muscle accretion and function, and combat sarcopenia in the elderly. Further research is needed to expand our knowledge about the mechanisms and the possible treatments that can maintain muscle function and prevent age-related loss of function and falls.

## Figures and Tables

**Figure 1 nutrients-15-04073-f001:**
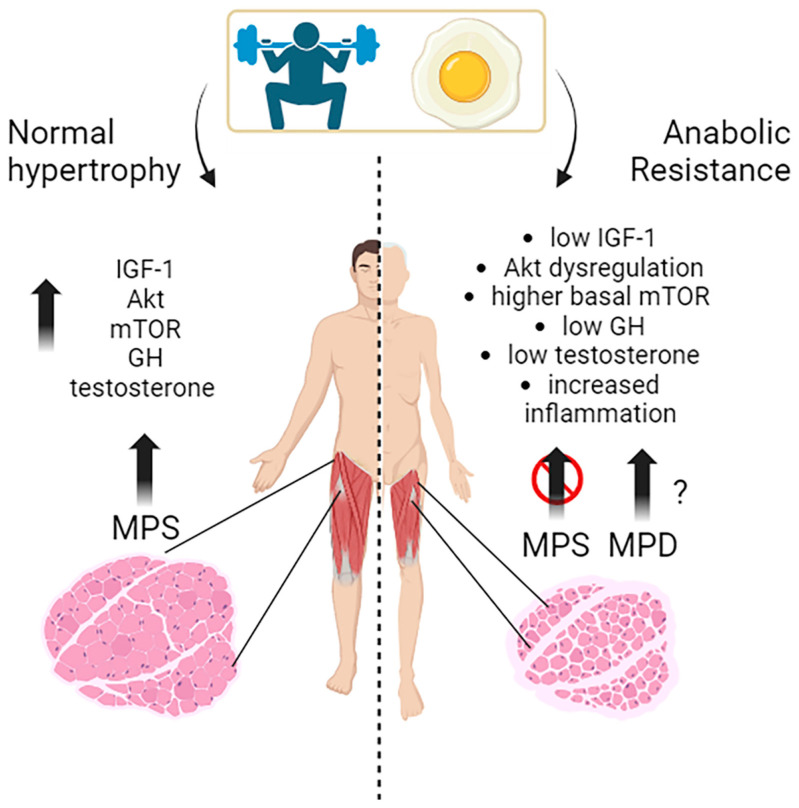
Here is a diagram showing the typical triggers for muscle growth in young people, which include exercise and high-quality protein intake. These stimuli activate various molecular pathways that lead to muscle hypertrophy. However, on the right-hand side of the diagram, you can see that the same stimuli do not have the same effect on skeletal muscle in cases of anabolic resistance.

**Figure 2 nutrients-15-04073-f002:**
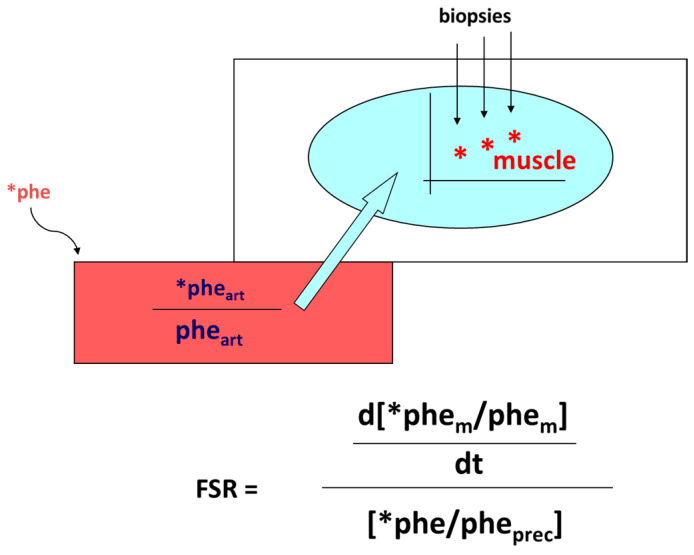
The figure schematically illustrates the methodology commonly employed to determine skeletal muscle fractional protein synthesis rate (FSR) with the combined infusion of an amino acid tracer (in this example, phenylalanine, (phe)) and its timed incorporation into skeletal muscle (measured by biopsy). Absolute Muscle Protein Synthesis (MPS) rate is calculated as the product of FSR times muscle protein mass. The asterisks (*) indicate the labeled amino acid. Abbreviations used in the Figure: art: artery; m: muscle; prec.: the enrichment (or the specific activity) of the “precursor” (i.e., the labeled amino acid), used in the calculations.

**Figure 3 nutrients-15-04073-f003:**
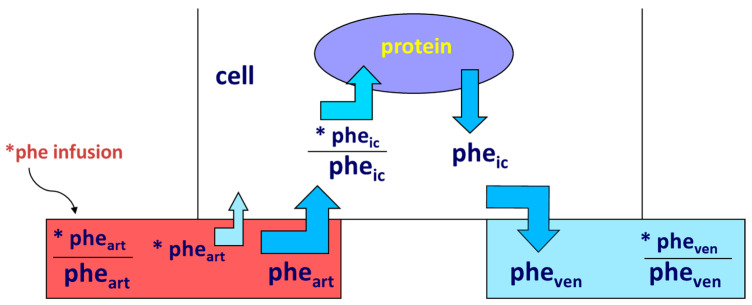
Measurement of skeletal muscle protein synthesis and degradation with arterial/venous combined with isotope infusion. The figure schematically depicts the measurements performed in the perfusing artery and in the deep vein draining blood from the sampled muscular-rich district (i.e., the leg or the forearm). In this example, the indicator essential amino acid is phenylalanine (Phe), which is utilized by muscle only for protein synthesis and is released from muscle only from protein degradation. Blood flow across the muscle district has to be measured too. Thus, from the measured phenylalanine utilization and release, it is possible to extrapolate protein degradation and synthesis. The asterisk (*) indicates the labeled amino acid. Abbreviations used in the Figure: art: artery; ven: vein; ic: intracellular. The equations used in the calculations are not reported here.

**Table 3 nutrients-15-04073-t003:** Nutritional indications of protein intake to optimize skeletal muscle mass and function, and combat anabolic resistance in ageing.

1. Consume a balanced diet rich in natural foods providing sufficient amounts of high-quality protein, to sustain (skeletal) muscle protein accretion;
2. Keep/increase protein intake (preferentially of high quality) at ≥1.5 g/kg/day;
3. Provide ≥ 30 g protein at each main meal, equally spread into the meals, to promote an optimal *per meal* stimulation of MPS;
4. Prefer protein-rich natural foods over protein-rich supplements;
5. Avoid as far as possible continuous 24 h nutrition;
6. Provide sufficient energy;
7. When adding supplements:
a. Use them only in specific cases;
b. Use EAA supplements preferentially rich in the BCAA;
c. Add leucine either to natural protein foods or to the AA supplements;
d. Consider that intake of protein supplements can proportionally lead to an involuntary reduction in the intake of protein-rich natural foods, thus partially offsetting their anabolic effect;
8. When combined with exercise, assume nutrition taking into account:
a. The timing of administration:
i. pre-exercise;
ii. at the beginning (t = 0′) (preferred);
iii. or following exercise…
b. Amount and type of nutrition?
i. Natural protein-rich foods? And/or:
ii. Supplements;
9. Consider other non-protein supplements:
1. Creatine, PUFA (Polyunsaturated fatty acids).

**Table 4 nutrients-15-04073-t004:** List and quantities of common animal and vegetal foods providing ≈ 30 g protein, and their caloric content, useful to improve muscle anabolism in ageing with exercise.

**Animal Foods**	**g**	**Calories**
Cow whole milk	909	582
Ricotta from cow milk	340	498
Ricotta from buffalo milk	286	606
Low-fat, fresh cheese	98	446
High-fat, aged cheese	89	347
Chicken egg (*n* = 4.4)	242	322
Beef meat	136	151
Pork meat	183	268
Chicken breast	131	131
Bresaola	94	142
Turkey breast	125	134
Seabass (filets)	141	210
Fresh salmon	163	302
**Vegetal Foods**	**g**	**Calories**
Soybeans	100	404
Wheat meal	273	927
Corn meal	345	1248
Beans	566	752
Quinoa	234	863
Oatmeal	238	917
Rice	450	1487
Lentils	475	436
Chickpeas	429	514

## Data Availability

Not applicable.

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
