# Peer review of "Anabolic Resistance in the Pathogenesis of Sarcopenia in the Elderly: Role of Nutrition and Exercise in Young and Old People"

_nutrients, 2023, doi:10.3390/nu15184073_

Round 1

Reviewer 1 Report

Comments to the Authors of manuscript ID: nutrients-2566572 entitled “Anabolic resistance in the pathogenesis of sarcopenia in the elderly.”.

It seems that the Authors have become confused in their thought process. Certain parts of the text are unrelated to the content and do not align with the stated title of the work.

1. Introduction lacks the basal definition of sarcopenia. In common use, there are 3 separate definitions established by 3 different entities.

2. the stages also should be mentioned including dynapenia.

3. the description of epidemiology is needed.

4. It should be mentioned that muscle fiber II type are reduced in older person

5. It lacks risk factors

6. Table 1 – what is link between the sarcopenia in the elderly and young adults? This table is useless.

7. L 274- generally this section (2.a) should be rephrased. In this form it does not fit taking into account the title

8. L 302-310 – it does not fit

9. L 334 - what newborn pigs have to do with sarcopenia in the elderly? It is off topic

10. part 3 – it should be explained that sarcopenia is primary and secondary, and then it has any sens to present the effect of glucocorticoids

11. 3.c. So far, it has not been precisely determined which genes are directly related to the possibility of developing sarcopenia. The studies that have been carried out so far have qualified genes predisposing to such properties that showed a correlation with muscle mass, muscle building or impact on their functionality to any extent. Cell cultures or animal models were used for this purpose. However, when IGF-1 is mentioned, the role of some genes (MSTN, CNTF, CNTFR, AR, IGF ect…) should be mentioned.

Author Response

Dear reviewer,

Thank you for your valuable suggestions. Enclosed, please find our response point by point.

Reviewer 2 Report

The authors offer an excellent manuscript on sarcopenia.

I particularly agree with the practical tips given, for example, increasing protein intake is very important.

There are a few points that I would consider better:

- The manuscript is more focused on the nutritional aspect, I would underline this from the title.

- I would try to expand the section relating to physical exercise, characterizing the various types and emphasizing how alternatives may also be possible to the classic schemes, also for greater compliance (which with age is not certain )

- I would consider the action of the muscle IGF1 isoform or MGF that is evoked by strength exercise

It needs some revision

Author Response

(The authors gave the same response as above.)

Round 2

Reviewer 1 Report

The authors' revisions were not substantial enough for the paper to be considered for publication.

1.       Introduction lacks the basal definition of sarcopenia. In common use, there are 3 separate definitions established by 3 different entities.

The reader who opens the publication should possess essential information, as not everyone will read the entire SI to acquire this knowledge.

2.       The description of epidemiology is needed.

one sentence about the percentage of sarcopenia among elderly people will not lengthen a meaningful text.

3.       Table 1 – what is link between the sarcopenia in the elderly and young adults? This table is useless.

This table should me removed especially that the title inorms that the paper is about old people

4.       L 274- generally this section (2.a) should be rephrased. In this form it does not fit taking into account the title.

This section does not fit due to it speaks about young or middle-age men. There are substantial difference between old and middle-age or young people, for this reason the discussion in this section is lessuse.

 Physiological processes exhibit notable variations between older and younger individuals due to the effects of aging. As people age, their bodily systems gradually undergo changes that can impact various functions. Some key differences include:

Metabolism: Metabolic rate tends to decrease with age, resulting in a slower rate of energy expenditure. This can lead to weight gain and increased susceptibility to obesity-related conditions.

Muscle Mass: the reduction in mascle mass affects mobility and overall physical function. Muscles contribute less to energy expenditure 

Bone Density: Bone density tends to decrease with age, making older individuals more prone to fractures and osteoporosis.

Cardiovascular System: Aging can lead to changes in blood vessels, contributing to decreased elasticity and increased risk of hypertension and cardiovascular diseases.

Hormonal Changes: Hormone levels, such as estrogen and testosterone, decrease with age, impacting reproductive function, sexual health, and hormones involving in the gut-brain axis.

Immune System: The immune system's efficiency can decline, resulting in a higher susceptibility to infections and a diminished response to vaccinations.

Cognitive abilities may change with age as well as vision and hearing.

Digestive System: Digestive functions can be affected by a decrease in stomach acid production and changes in bowel movements.

Sleep Patterns: Older individuals may experience changes in sleep patterns, including difficulties falling asleep and staying asleep.

Respiratory Function: Lung capacity and respiratory function may decrease, affecting overall lung health and physical endurance.

 Differences in digestive function in particular are the main reason why this section is wrong. Otherwise, young people will use all the supplements.

 5.       L 358- 370 (previously 302-310) – it does not fit

It does not fit due to the same reason as above.

Author Response

Dear Reviewer,
